# Genome-wide CRISPR screen identifies protein pathways modulating tau protein levels in neurons

Carlos G. Sanchez[1], Christopher M. Acker[1], Audrey Gray[1], Malini Varadarajan[2], Cheng Song[2], Nadire R. Cochran[2], Steven Paula[2], Alicia Lindeman[2], Shaojian An [2], Gregory McAllister[2], John Alford[2], John Reece-Hoyes[2], Carsten Russ[2], Lucas Craig[1], Ketthsy Capre[1], Christian Doherty[1], Gregory R. Hoffman[2], Sarah J. Luchansky[2,3], Manuela Polydoro[1], Ricardo Dolmetsch[1] & Fiona Elwood [1✉]

Aggregates of hyperphosphorylated tau protein are a pathological hallmark of more than 20 distinct neurodegenerative diseases, including Alzheimer's disease, progressive supranuclear palsy, and frontotemporal dementia. While the exact mechanism of tau aggregation is unknown, the accumulation of aggregates correlates with disease progression. Here we report a genome-wide CRISPR screen to identify modulators of endogenous tau protein for the first time. Primary screens performed in SH-SY5Y cells, identified positive and negative regulators of tau protein levels. Hit validation of the top 43 candidate genes was performed using Ngn2-induced human cortical excitatory neurons. Using this approach, genes and pathways involved in modulation of endogenous tau levels were identified, including chromatin modifying enzymes, neddylation and ubiquitin pathway members, and components of the mTOR pathway. TSC1, a critical component of the mTOR pathway, was further validated in vivo, demonstrating the relevance of this screening strategy. These findings may have implications for treating neurodegenerative diseases in the future.

[1] Department of Neuroscience, Novartis Institutes for BioMedical Research, Cambridge, MA, USA. [2] Department of Chemical Biology and Therapeutics, Novartis Institutes for BioMedical Research, Cambridge, MA, USA. [3]Present address: Vertex Pharmaceuticals, Boston, MA, USA. ✉email: fiona.elwood@novartis.com

Tau protein is encoded by the *MAPT* gene on chromosome 17 and is predominantly expressed in the axons of neurons. This soluble and natively unstructured protein normally functions to aid in the polymerization and stabilization of microtubules. Six isoforms of tau are expressed in the central nervous system (CNS), and vary by the inclusion or exclusion of exons 2 and 3 that encode the N-terminus of the protein (0N, 1N, and 2N), and exon 10 that encodes an additional microtubule binding domain repeat towards the C-terminus of the protein (3R and 4R). In disease, tau is extensively post-translationally modified, detaches from microtubules and aggregates to form multiple species of oligomers, fibrils, and ultimately paired helical filaments that form neurofibrillary tangles (NFTs)[1,2]. Aggregates of hyperphosphorylated tau protein are the hallmarks of many neurodegenerative diseases. "Tauopathies" can be classified as either "primary", where the tau lesion is upstream in the pathogenic cascade and tau is the predominant aggregated protein found in the brains of patients, these include progressive supranuclear palsy (PSP), corticobasal degeneration (CBD), Pick's disease (PiD), and frontotemporal dementia with parkinsonism-17 (FTDP-17); or "secondary" tauopathies, where the tau lesion is downstream of a causative insult such as Aβ amyloid plaques in Alzheimer's disease (AD)[3,4]. In AD patients, Aβ amyloid pathology plays a substantial role in initiating the disease process, but the formation of tau NFTs exacerbate neurodegeneration leading to dementia in these patients[5].

Tau pathology can cause neurodegeneration independent of Aβ in patients suffering from primary tauopathies. *MAPT* mutations are present in approximately 5% patients with FTDP-17[6–9]. While aggregated, extensively post-translationally modified species of tau are present in diseased brains, the exact forms that are the most toxic remain a matter of debate. Interestingly, reducing or ablating endogenous tau levels in mice is well tolerated[10–13], and therefore several therapeutic approaches that aim to reduce total tau levels have been pursued[14]. These include targeting the mRNA with an antisense oligonucleotide (ASO)[14] and using zinc finger proteins and gene therapy to target tau expression levels[15]. Other approaches have attempted to target extracellular tau using passive or active immunotherapy (reviewed in the ref. [16]), or drive degradation of aggregated tau by activation of the proteasome pathway[17,18]. These novel modalities are at various phases of drug development from early preclinical testing to phase II clinical trials[16]. Because of the high levels of tau molecular diversity in AD[19,20], one major advantage of tau lowering approaches, is that they can potentially reduce all species of tau proteins, including post translationally modified tau species that cause pathology. However, one disadvantage of the current tau therapies is that they rely predominantly on large molecules, including antibodies and ASOs that have poor access to the brain. Therefore, there is great interest in identifying targets that regulate the expression or degradation of tau, and can be modulated with small brain penetrant molecules to reduce pathogenic tau in patients.

In order to identify protein pathways modulating endogenous tau protein levels, we employed an unbiased high throughput CRISPR pooled screening strategy[21] using an SH-SY5Y neuroblastoma cell line. Several candidate genes were then validated in human excitatory neurons. Additional in vivo validation of tuberous sclerosis protein 1 (TSC1), a critical component of the mTOR pathway, demonstrated that negatively regulates tau levels in neurons and in the brain. Hence, we provide a useful resource that has identified pathways and candidate genes that modulate the overall levels of tau protein in neurons, and could help in future identification of novel therapeutics for AD.

## Results

**Validation of SH-SY5Y neuroblastoma line for tau CRISPR screen.** To identify modulators of tau protein levels we employed a high throughput CRISPR pooled screening strategy to identify genes that, when knocked out, either increased or decreased expression of endogenous tau (Fig. 1a; see ref. [21]). Two SH-SY5Y neuroblastoma cell lines that endogenously express total 3R tau, and stably expressed FLAG-Cas9 were used (Fig. 1b, c, Supplementary Figs. 1a, e and 5b–d). The first, being a pooled cell line, hereafter referred as PC1; and the second being a single clone expanded from the PC1, hereafter referred as SC2. The efficiency of Cas9 editing in both PC1 and SC2 was evaluated. Two guide RNAs (gRNAs) that targeted *MAPT* (the tau gene), were introduced to the cells via lentiviral transduction. To measure Cas9 editing efficiency and subsequent tau reduction, western blot analysis was performed for tau protein levels from lysates of PC1 and SC2 cells. A significant reduction of tau protein in both PC1 and SC2 cells could be seen at 14 days of post-infection; and this reduction was maintained at 21 and 30 days of post gRNA infection (Fig. 1c, Supplementary Figs. 1a and 5a). On average, 85% or 75% reduction of tau was maintained over the time period examined with gRNA *MAPT*-1 and *MAPT*-2, respectively. In addition, immunocytochemistry (ICC) was performed using a total tau antibody, SP70. Fifty percent reduction of tau protein was observed as early as 7 days of post gRNA infection (Fig. 1d). FACS can be used to differentiate populations of cells with low or high protein levels[21]; therefore we determined if SH-SY5Y cells could be stained for endogenous tau and sorted based on tau levels. Two gRNAs against *MAPT* were selected, and PC1 cells were edited for 14 days. After editing, PC1 cells were fixed, stained using the total tau antibody SP70, and FACS sorted. Edited cells had lower FITC fluorescence compared to controls (Supplementary Fig. 1b, c), indicating reduced tau protein levels. This indicated that expression of endogenous tau protein in SH-SY5Y cells can be modulated using gRNAs and stable expression of Cas9. In addition, FACS can be used to differentiate and sort cells with reduced endogenous tau levels.

**Genome wide CRISPR screen in SH-SY5Y neuroblastoma cell line.** To perform a genome-wide CRISPR screen in search of modulators of tau protein levels, cells from the pooled cell line (PC1) were infected with a pooled lentiviral gRNA library encoding 90,000 individual guides targeting 18,360 human genes (five gRNA per gene, see refs. [22–26]). After infection, cells with stable gRNA integration were selected, and edited for 21 or 30 days. These time points were selected because previous studies reported the half-life of tau protein between 6–23 days[27], therefore, we wanted to allow ample time for the protein being knocked down to effect tau protein levels. After editing, cells were collected, fixed, stained with a total tau antibody, and FACS sorted to distinguish cell populations with low tau expression (low quartile 25%) and high tau expression (high quartile 25%; Fig. 1a). An additional population of cells were left unsorted, and the gRNAs were sequenced to determine the distribution of gRNAs present in the entire cell population after editing. Genes that were depleted in the unsorted population compared to the input library could have impacted viability or proliferation. This analysis of unsorted cells served as a counter screen and genes identified in both screening time points (21 or 30 days) by this method were excluded from subsequent analysis (Supplementary Data 1). After sorting, genomic DNA was isolated from the low and high tau cell populations and analyzed by high-throughput

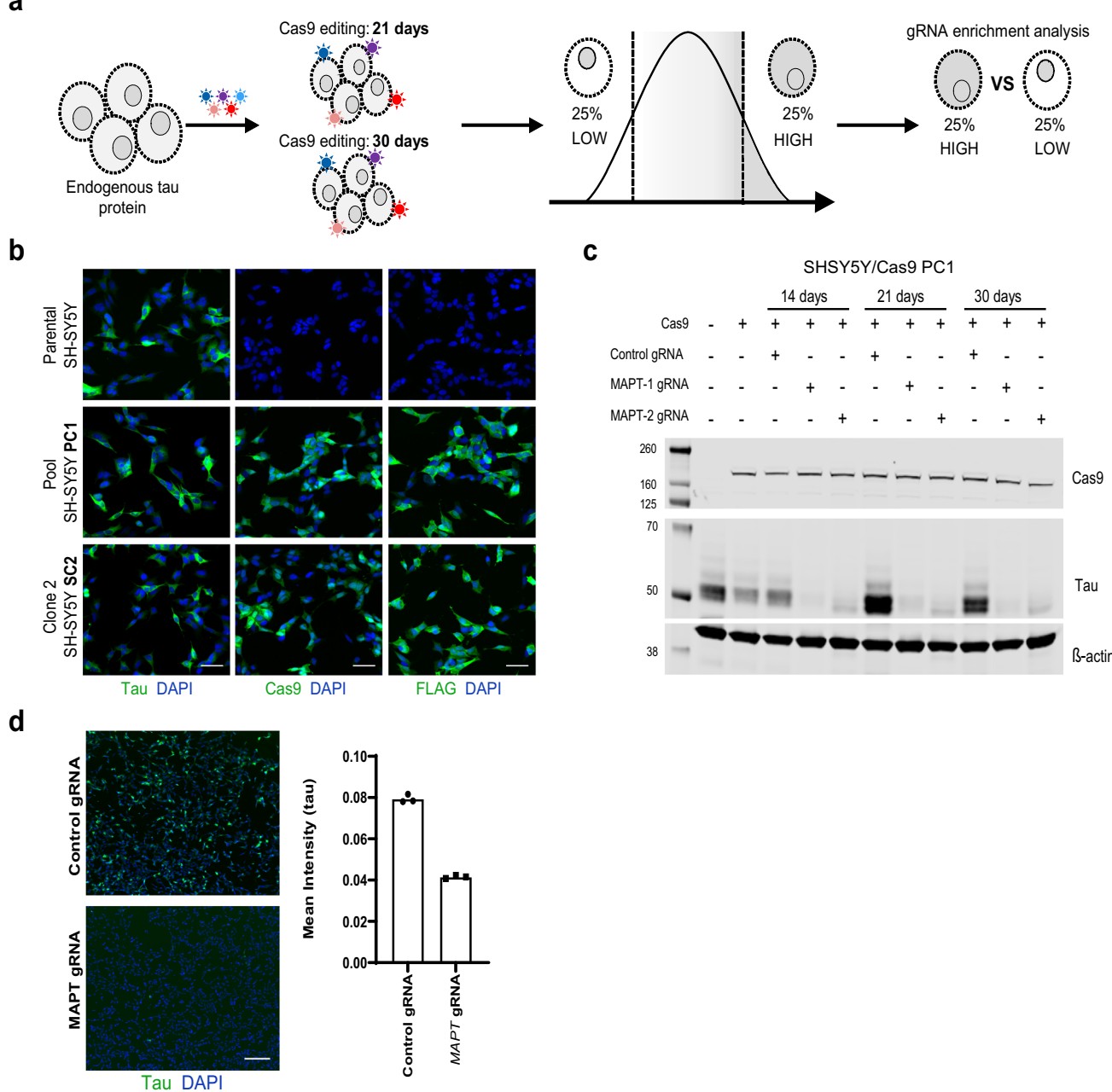

**Fig. 1 Characterization of SH-SHY5Y cell lines expressing Cas9. a** Schematic representation of primary genome-wide screen performed in SH-SY5Y cells expressing Cas9. **b** Immunocytochemistry of endogenous tau (SP70), Cas9 and FLAG proteins in parental and stable cell lines expressing Cas9. Pooled cells (PC1) and single clones (SC2) are depicted. Scale bar represents 50 μm. **c** Tau (SP70) and Cas9 western blot analysis on SH-SH5Y/Cas9 (PC1) cells infected with control or *MAPT* gRNAs for 14, 21, or 30 days of editing. **d** Immunocytochemistry of endogenous tau in SH-SH5Y/Cas9 (PC1) cells infected with control or *MAPT* gRNAs for 7 days of editing. Scale bars represent 200 μm. Data is represented as mean fluorescence intensity ± SEM. Control is a non-target gRNA.

sequencing to determine the enrichment of gRNA sequences in each population of cells. gRNAs enriched in either low or high tau expressing populations are expected to be specific for negative or positive modulators of tau protein levels, respectively.

To determine statistical significance of gRNA enrichment we performed a redundant siRNA activity (RSA; −log *p*-value) analysis[28], and determined fold change of enrichment (robust *z*-score) of gRNAs in the 25% cells with high tau levels vs. low tau levels. RSA down scores were used to find genes enriched in the 25% of cells with the lowest tau protein, and RSA up scores were used to find genes enriched in the 25% of cells with highest tau

protein. Each gRNA demonstrated a different editing efficiency and thus had variable effects on tau levels. For some genes, all the gRNAs had a similar effect, whereas for others there were clearly gRNAs that were outliers. To identify genes that had the greatest magnitude of effect on impacting tau levels, the fold-change enrichment of specific gRNAs (magnitude of effect) was compared to the RSA score for all the gRNAs for each gene (significance). The gRNAs for each gene were ranked based on the fold-change robust *z*-score for each guide. To reduce the influence of outlying gRNAs, the second ranked gRNA (closest to Q1 in the distribution of the gRNAs) and the fourth ranked

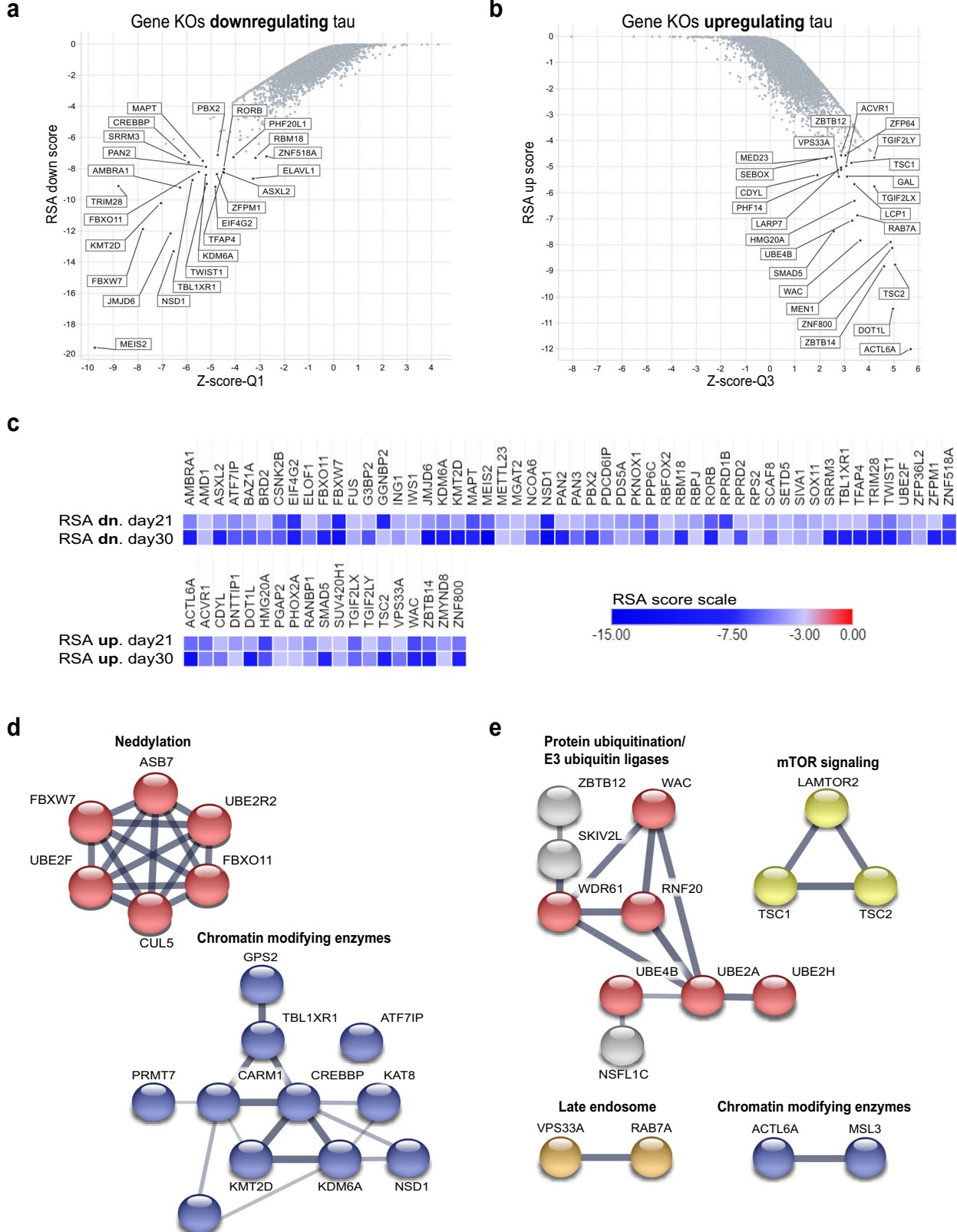

**Fig. 2 Gene knockouts upregulating and downregulating tau protein in SH-SY5Y cells after CRISPR editing. a**, **b** Scatter plots with RSA scores (*Y*-axis) vs. *z*-score quartiles (*X*-axis), for 30 day edited group from primary genome-wide screen. Plots shows genes downregulating (**a**) and upregulating (**b**) tau protein levels. Highly significant candidate genes are labeled. **c** Heat maps representing candidate genes with an RSA score ≤−3.0 in primary genome-wide screens (21 and 30 day time points). **d**, **e** STRING enrichment pathways for highly validated genes (primary and mini array screens). Links between nodes are highly confident connections. **d** Pathways identified for genes downregulating tau protein levels. **e** Pathways identified for genes upregulating tau protein levels. KO knockout, Q quartile, Dn down, RSA redundant siRNA activity score.

gRNA (closest to Q3 in the distribution of the gRNAs) were used for the comparison to RSA scores (Supplementary Fig. 1d). To identify genes that significantly reduced tau, the guide with the second lowest z-score (Q1 fold-change robust z-score, X-axis) was compared to the RSA-down score (Y-axis) for each gene (Fig. 2a, Supplementary Fig. 2a, and Supplementary Data 2; see "Methods" section). In contrast, to identify genes that significantly increased tau, RSA-up scores (Y-axis) were compared to quartile 3 (Q3) fold change robust z-scores (X-axis). (Fig. 2b, Supplementary Fig. 2b, and Supplementary Data 2; see "Methods" section).

**Distribution of *MAPT* gRNAs validated screen**. To confirm that efficient editing had occurred in the genome-wide screen, the five gRNAs targeting *MAPT* at both editing time points of each of the primary screens were reviewed. In the previous experiments, Cas9-SH-SY5Y cells targeted with *MAPT* gRNAs were consistently depleted in cells with highest tau levels and enriched in cells with the lowest tau levels (Fig. 2a, c and Supplementary Fig. 2a), therefore we expected to see a similar distribution of gRNAs in the primary screens. Indeed, *MAPT* gRNAs had a significant RSA score of <−6 and <−7, for the 21 and 30 day editing time points, respectively (Supplementary Data 2). This indicated that the screen had the capacity to identify gRNAs that alter the expression of endogenous tau levels.

**Identification of genes modulating tau protein levels**. The genome-wide screen was used to identify genes regulating the levels of tau protein. Using an RSA-down and RSA-up cut off score of <−3; candidate gene lists were compiled for gene knockouts downregulating and upregulating tau protein levels at 21 days and 30 days of editing. Fifty-two genes were identified that decreased endogenous tau protein levels at both timepoints (Fig. 2c and Supplementary Data 2). In addition, 19 genes were identified that increased endogenous tau protein levels at both timepoints (Fig. 2c and Supplementary Data 2). Genes that could impact viability or proliferation were excluded prior to the enrichment analysis, thereby removing potential genes with confounding mechanisms of action for reducing tau levels from the hit list (Supplementary Data 1). Some of the genes down regulating tau protein levels at both timepoints, include bromodomain containing protein 2 (BRD2), RNA-binding protein fused in sarcoma (FUS), and tripartite motif containing 28 (TRIM28; Fig. 2c). Reduction of TRIM28 in adult mice has previously been reported to lower tau levels[29,30], increasing confidence in the screening results. Genes up regulating tau protein at both timepoints included, paired like homeobox 2A (PHOX2A), tuberous sclerosis 2 protein (TSC2), and vacuolar protein sorting 33A (VPS33A). Using this pooled strategy, we performed two unbiased genome-wide primary screens that uncovered genes regulating tau protein levels in neuroblastoma cells.

**Validation of hits using a mini pool CRISPR screen**. Endogenous tau levels are tightly regulated and did not change by more than 2-fold in the genome-wide primary screens (Supplementary Data 2). This narrow window raised the possibility that the screen would identify false positives, and therefore stringent validation assays were performed. To validate the screening results, a secondary validation screen was performed using a custom mini pool library. This library contained ten gRNAs for every top candidate gene with an RSA down/up score of <−3 in both timepoints of the primary screen, totaling 481 genes. These included 266 gene knockouts that decreased tau protein levels and 215 that increased tau protein levels. Additional control genes were included for a total of 5185 gRNA elements. Validation screens were performed on two independent cell lines, the pooled

PC1 cell line (used in primary screens), and a cell line derived from a single clone of PC1, the SC2 cell line. These two cell lines were used to make sure results could be recapitulated in different cellular contexts. Since both cell lines showed nearly complete tau protein knockdown after 14 days of editing using *MAPT* gRNAs (Fig. 1c and Supplementary Fig. 1a), and to address the possibility that influences on tau protein levels could be an indirect effect, an additional 14 day editing time point was added to the validation screens. Therefore, cells were edited for 14, 21, and 29 days (Supplementary Fig. 3a). Validated genes were defined as hits in at least two of the six mini pool validation screens, with a stringent RSA score cut-off of ≤−5 (Supplementary Fig. 3b, c and Supplementary Data 3). 93 of 266 (35%) gene knockouts that reduced tau levels in the primary screens were confirmed in the secondary mini pool screens. 62 of 215 (29%) gene knockouts that increased tau levels in the primary screens were confirmed in the secondary mini pool screens.

**Network analysis identified protein pathways modulating tau protein levels**. Primary genome-wide screens identified 481 genes regulating tau levels in SH-SY5Y cells, from these 93 were validated as genes that reduced tau levels and 62 were validated as genes that increased tau levels. To identify if there were specific pathways modulating tau protein levels, an in-depth STRING[31] and enrichment analysis was performed using the 155 validated genes from the secondary mini pool screens. STRING combines known physical/functional protein associations, computational predictions, and known protein interactions aggregated from primary databases, to predict protein–protein interactions that form protein networks. The primary databases include Reactome (RCTM) composed of peer reviewed and manually curated pathway analysis[32], Gene Ontology (GO), which includes gene annotations for molecular function, biological process and cellular component, and the Kyoto Encyclopedia of Genes and Genomes (KEGG), which uses known datasets from genome sequencing and other high-throughput experimental technologies[33].

When performing the analysis on the validated tau down regulating genes, RCTM pathway analysis uncovered that chromatin modifying enzymes and proteins involved in neddylation, ubiquitination, and proteasome degradation played important roles in regulating tau protein levels (Fig. 2d blue-red, Supplementary Data 4). In addition, GO analysis revealed that proteins playing roles in RNA metabolism may also be regulating tau levels (Supplementary Fig. 2c—green and Supplementary Data 4). Interestingly, disruption of only two of the seven cullin–RING ligase complexes decreased tau levels in SHSY-5Y cells. Knockout of FBXW7 and FBXO11 affects the SKP1-Cullin1-F-box complex and knockout of Cul5 affects the Elongin-Cullin5-SOCS-box complex (Table 1, see ref. [34]). These complexes are also directly regulated by another hit, SENP8, a deneddylase that regulates optimal neddylation of cullin proteins to stabilize substrates for ubiquitination (Supplementary Fig. 3b)[35]. This indicates an involvement of these two cullin-RING ligase complexes and SENP8 in the regulation of tau protein levels, distinct from the role of other cullin-RING ligase complexes.

When performing the RCTM enrichment analysis on the validated tau up regulating genes, we found similar pathways as with our analysis on the tau down regulating genes, these include proteins involved in ubiquitination (E3 ligases) and chromatin modifying enzymes (Fig. 2e—red/blue and Supplementary Data 5). In addition, we uncovered several pathways of interest that increase tau levels, these include mammalian Target of Rapamycin (mTOR) signaling, endolysosomal membrane

**Table 1 CRL1 and CRL5 reduced tau protein levels.**

| Name | Cullin | Complex | Protein Hits | Phenotype |
|---|---|---|---|---|
| SCF (CRL1) | CUL1 | F-box protein/SKP1/CUL1/RING | FBXW7, FBXO11 | Tau reduction |
| CRL2 | CUL2 | SOCS/BC-box protein-elongin-CUL2/RING | None | No phenotype |
| CRL3 | CUL3 | BTB-domain protein/CUL3/RING | None | No phenotype |
| CRL4A | CUL4A | DCAF/DDB1/CUL4A/RING | DCAF7 | Proliferation |
| CRL4B | CUL4B | DCAF/DDB1/CUL4BRING | DCAF7 | Proliferation |
| CRL5 | CUL5 | SOCS/BC-box protein-elongin/CUL5/RING | ASB7, UBE2F, Cul5 | Tau reduction |
| CRL7 | CUL7 | FBXW8/SKP1/CUL7/RING | None | No phenotype |

proteins (Fig. 2e—yellow/gold) and TGF-beta/BMP signaling (Supplementary Fig. 2d—light blue and Supplementary Data 5). Interestingly, our screen uncovered that knockouts of three E2 Ubiquitin-Conjugating Enzymes, UBE4B, UBE2A, and UBE2H increased tau levels. Overall our screening strategy identified protein networks that have specific roles in the regulation of proteins that cause neurodegenerative diseases.

**TSC1/2 increased tau protein levels in human neurons.** To confirm hits in a post mitotic, human, neuronal system, an H1 stem cell line that expressed the Neurogenin-2 (Ngn2) transcription factor under the control of the tetracycline trans-activator for rapid induction of human excitatory neurons was employed, referred hereafter as iNgn2 neurons[36,37]. At 14 days of post-differentiation, iNgn2 neurons express endogenous tau protein and constitutively express Cas9 to enable CRISPR editing (Fig. 3a). In addition, iNgn2 neurons express neuronal markers TUBB1 and MAP2, and do not express stem cell marker OCT4, consistent with previous studies (Supplementary Fig. 4a, see ref. [37]). However, at 14 days of post-differentiation, iNgn2 neurons only express the 3R tau isoform and lack expression of the 4R tau isoform (Supplementary Figs. 4b and 5b–d). iNgn2 neurons were plated three days after doxycycline induction of Ngn2. On day 5, iNgn2 neurons were infected with lentivirus containing a specific gRNA targeting a gene of interest. Editing in differentiating iNgn2 neurons occurred over 14 days and neurons were lysed on day 20. Neurons were observed prior to lysis to assess viability, and samples were normalized to total protein levels to account for any toxicity caused by editing. Tau protein levels were measured using an AlphaLISA using total tau antibodies BT2 and HT7 (Fig. 3b). Genes that had an RSA score of $<-5$ in at least six of the eight total SH-SY5Y CRISPR screens performed were selected for neuronal validation. This included 20 gene knockouts that reduced tau protein levels, and 23 gene knockouts that increased tau protein levels in the primary and secondary screens (Fig. 3c and Supplementary Data 6). The top two performing gRNAs for each of the genes (gene knockouts with the lowest and highest fold change for tau down and tau up regulators, respectively) were selected for human neuronal validation (Supplementary Fig. 4c, d and Supplementary Data 7 and 8). In addition, 2 gRNAs for *MAPT* and a non-targeting gRNA were selected as controls. Guides were packaged and iNgn2 neurons were infected and selected for guide integration. The fold-change of tau protein levels after specific gene knockouts was normalized to non-targeting controls (Supplementary Fig. 4e, f). This analysis revealed that genes down regulating tau levels in the primary and validation screens also trended toward reducing tau protein levels in iNgn2 neurons (Fig. 3d—blue bars). On the other hand, genes selected for up regulating tau levels in primary and validation screens trended toward increasing tau protein levels in iNgn2 neurons (Fig. 3d—green bars).

To determine if candidate genes from the screens performed in SH-SY5Y cells further validated in a human neuronal model,

*z*-scores were generated for each of the two gRNAs tested per gene. To consider a gene validated in neurons, both gRNAs targeting the gene had to show the same phenotype observed in screens performed in SH-SY5Y cells. Following these criteria, knockouts of Eukaryotic Translation Initiation Factor 4 Gamma 2 (EIF4G2), Myelin Transcription Factor 1 (MYT1), and Nuclear Receptor Binding SET Domain Protein 1 (NSD1) further validated in neuronal iNgn2 cultures by reducing tau levels at least one standard deviation below the mean. Indeed, the chromatin modifying enzyme NSD1, a member of one of the protein networks identified by STRING analysis, significantly decreased tau protein levels in neuronal cultures (Figs. 2d—blue and Fig. 3e). To our surprise, knockouts of F-Box and WD Repeat Domain Containing 7 (FBXW7) and Nuclear Receptor Coactivator 6 (NCOA6), originally selected for decreasing tau levels in SH-SY5Y cells, increased tau levels in iNgn2 neurons by almost two standard deviations above the mean (Fig. 3e). gRNAs targeting Tuberous Sclerosis 1 and 2 (TSC1/TSC2) both increased tau levels in iNgn2 neuronal cultures (Fig. 3e); further validating one of the upregulation networks, the mTOR signaling pathway (Fig. 2e—yellow). By performing genome-wide screens, retrieving candidate genes, and validating these genes in neuronal cultures, several protein networks that regulate tau levels in both neuroblastoma cells and differentiated human iNgn2 neurons were validated.

**Tsc1 knockout increased tau protein levels in vivo.** TSC1 and TSC2 form the GTPase TSC complex that inhibits RAS homolog enriched in brain (RHEB). In normal cell conditions RHEB activates the mTOR pathway causing mRNA translation via phosphorylation of S6 kinase and the eukaryotic initiation factor 4E binding protein (4EBP1). Cellular homeostasis is maintained via adenosine monophosphate-activated protein kinase (AMPK), which inactivates the TSC complex leading to the activation of RHEB, therefore mutations inhibiting TSC1/2 lead to increases in mTOR activity[38,39].

To determine if the genetic modifiers of tau levels identified in the screen could also modify tau levels in vivo, TCS1, a candidate gene identified from primary and validation screens in neuro-blastoma cells, and further validated in neurons, was selected for in vivo validation. For this a neuronal Tsc1 cKO mouse model of tuberous sclerosis was characterized to determine the effect on tau protein levels in the brain. This model used a CaMKII promoter to drive Cre recombination-dependent *Tsc1* knockout in neurons and show elevated levels of mTOR signaling[40]. Forebrains were dissected from postnatal mice at day 30 and day 38, proteins were extracted and protein levels measured using quantitative western blot analysis. Consistent with published data, we observed an age dependent increase of pS6 levels in the brain of Tsc1 cKO mice compared to control animals (Fig. 4a) indicating elevated mTOR signaling. In parallel, a significant increase of tau protein was observed in the brain of Tsc1 cKO mice (Fig. 4b). Here, we confirmed the increase of tau protein by conditionally knocking

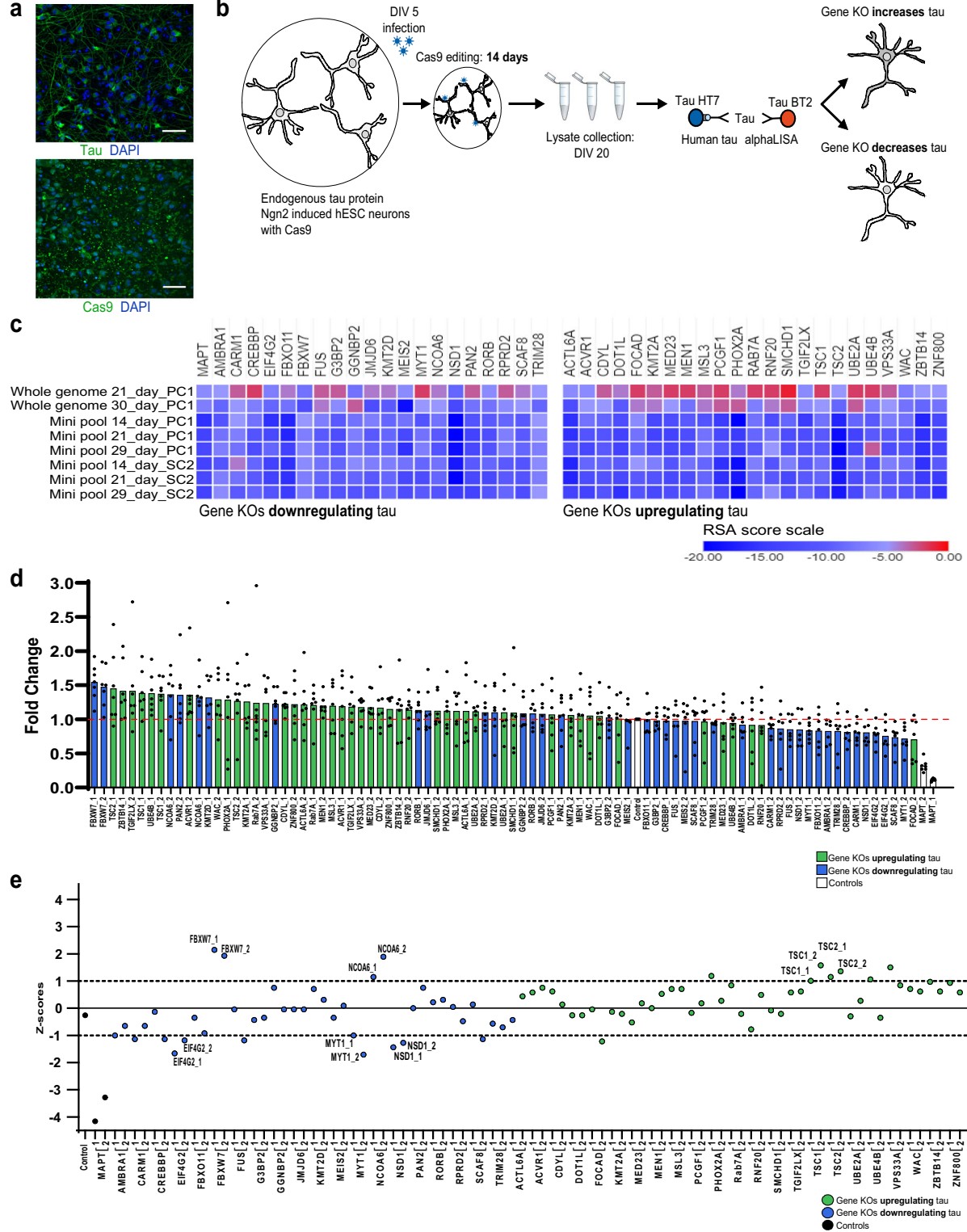

**Fig. 3 Candidate genes validated in a human neurons. a** Immunocytochemistry of endogenous tau (SP70) and Cas9 in iNgn2 neuronal cultures 14 days of post-differentiation. Scale bar represents 1000 μm. **b** Schematic representation of secondary well-based screen in human neurons. **c** Heat maps showing top 44 genes candidate genes with an RSA score ≤−5.0. Twenty-one downregulating genes (left) and 23 upregulating genes (right). **d** Quantitative analysis of tau protein levels in neurons following treatment with individual gRNAs. Gene knockouts downregulating tau (blue) and gene knockouts upregulating tau (green). Data is represented as fold change of tau protein levels compared to non-targeting control gRNAs ± SEM for seven biological replicates. **e** Z-score analysis on fold change for individually tested gRNAs, for gene knockouts downregulating tau (blue) or gene knockouts upregulating tau (green). Validated genes 1 standard deviation above or below the mean are labeled. KO knockout, DIV days in vitro, hESC human embryonic stem cells, Control is a non-target gRNA.

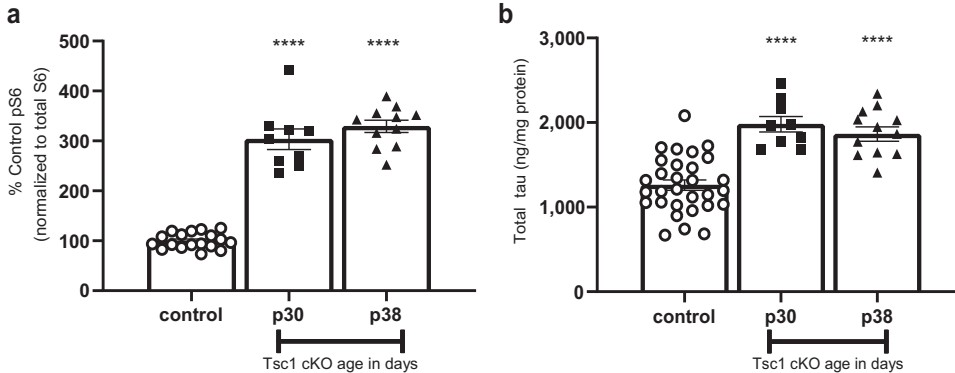

**Fig. 4 TSC1 knockout in mice increased tau protein levels. a** Phosphorylation of S6 (Serine 240/244) levels at postnatal day 30 (p30, squares) and postnatal day 38 (p38, triangles) compared to aged matched Cre- control mice (circles). Western blot data is represented as mean ± SEM, as the percentage of control animals. Each data point represents an individual animal. **b** Total tau levels at p30 (squares) and p38 (triangles) compared to aged matched Cre- control mice (circles). Data is represented as ng of tau protein/mg total protein ± SEM. One-way ANOVA was performed; ****$p < 0.0001$ as compared to control. Each data point represents an individual animal.

out Tsc1 in an in vivo model. These data demonstrated the utility of using this unbiased screening approach to identify genes modulating tau biology. Future studies will be essential to explore additional candidate genes and pathways identified from this genome-wide screen.

**Tsc1/2 knockdowns do not regulate tau aggregation in rat neurons**. Higher levels of tau protein via activation of mTOR signaling can lead to higher levels of phosphorylated tau, further increasing the amount if insoluble aggregated tau[41–43]. To determine if the TSC complex could also be playing a role in regulating tau aggregation levels, genes were evaluated in a rat neuronal aggregation assay[44]. Rat neurons were plated and siRNA pools targeting MAPT, Tsc1, Tsc2, and mTor were introduced at 5 days after plating. At day 7, tau seeds extracted from human Alzheimer's disease brains were introduced and neurons were incubated with tau seeds for 14 days. At day 21, cells were fixed with methanol to remove soluble tau protein and immunocytochemistry was performed using a rat specific tau antibody T49 and DAPI. Inclusions were quantified and then compared with non-targeting controls. Neurons incubated with siRNA pools for 14 days showed reduced gene expression; tau expression was reduced by 67%, Tsc1 by 39%, Tsc2 by 82%, and mTor by 65% (Supplementary Fig. 4g). We find that knockdown of Tsc1 and mTor gene expression, does not significantly increase or decrease tau aggregate counts in rat neurons incubated with tau seeds extracted from human Alzheimer's disease brains (Supplementary Fig. 4h, i). Surprisingly, knockdown of Tsc2 showed a slight significant decrease in tau aggregates when compared to non-target controls, however, this significance was not recapitulated when compared to PBS controls.

## Discussion
As our understanding of tau disease biology has evolved, so have the various strategies for targeting tau. Early tau-targeting therapeutic approaches focused on regulating intracellular aggregate formation by inhibiting tau kinases, inhibiting tau aggregation, or improving microtubule stabilization (reviewed in the ref. [16]). Recent tau-targeting therapeutic approaches have focused on binding extracellular tau to halt the propagation and spread of oligomeric tau aggregates, via passive or active immunotherapy (reviewed in the ref. [45]), or by implementing strategies to lower total tau levels[14,15]. As a potential alternate therapeutic strategy for reducing tau levels, or to further the understanding of novel

pathways that modulate the expression or degeneration of tau, we performed a high throughput FACS based genome-wide CRISPR screen in a neuroblastoma cell line that expresses endogenous tau protein. To our knowledge, this genome-wide screen is the first of its kind to identify modulators of endogenous tau protein levels. Several groups have attempted to identify modulators of tau transgenes using siRNA or shRNA modulation[29,46,47]. These screens face intrinsic challenges, including that hits may modulate the transgene but not endogenously regulated tau, and siRNA may result in partial knock down of a gene product or indeed show off-target activity reducing the expression of other genes. While measuring endogenous tau is more challenging from a technical perspective due to the relatively lower levels of protein expression, it is less likely to be influenced by experimental artifacts. Furthermore CRISPR editing results in complete knockout of gene products and less off-target activity[48].

SH-SY5Y cells are a human neuroblastoma cell line. While they show some neuronal characteristics including expression of neurotransmitters, they are different to human neurons in that they continue to divide, and do not develop elaborate neuronal processes and synaptic connections[49,50]. Genes and networks that modified tau levels in SH-SY5Y cells were identified and robustly validated in this study. The primary screens identified 266 genes decreasing, and 215 increasing tau protein levels, for a total of 481 candidate genes. Secondary screens further validated 93 of the 266 genes decreasing, and 62 of the 215 increasing tau levels. However only three of the top 20 tested genes reducing tau levels, and two of the top 23 tested genes increasing tau levels validated in human neurons. One of these hits (TSC1) was also validated in vivo, demonstrating the validity of this screening and triage approach. Remaining genes to be tested in neurons that meet our stringent criteria include 36 genes downregulating, and 14 genes upregulating tau protein levels in SH-SY5Y cells. These remaining genes are represented in the different protein networks identified through STRING analysis. The low validation rate in neurons may be due to the intrinsic differences between the SH-SY5Y cell line and Ngn2 human neurons, or it may reflect the fact that editing in human neurons occurred during differentiation of the neurons. While Ngn2 neurons offered the ability to test cell type specific phenotypes in a pure population of excitatory glutamatergic neurons without having to account for the possibility of a diluted phenotype from mixed neuronal cultures[37,51], the system has its limitations. At 14 days of post differentiation, the iNgn2 neurons do not express the 4R tau isoform and levels of total tau protein were not comparable to human brain levels. Previous

studies have attempted to address this Ngn2 limitation by culturing neurons for longer. This increased the levels of 4R tau, but the 3R:4R Tau expression ratio was not comparable to expression ratios in human brains[52]. Using an induced pluripotent stem cells (iPSC)-neuronal system from patients with tau mutations, differentiation for 150–365 days was required to see expression of multiple tau isoforms, yet these still did not compare to the levels seen in the human brains[53–55]. While a more robust human neuronal system with expression of both 3R and 4R tau isoforms would benefit studies looking at tau aggregation, the scope of this study is only looking at regulators of tau protein levels. For this purpose, the current neuronal system allowed for a rapid additional hit validation in a human post-mitotic neuronal system that expressed endogenous tau levels, and triage of hits for validation in a more mature in-vivo neuronal system. Additional approaches such as CRISPR-i or pharmacological inhibition in differentiated human neurons[52], or in vivo experiments may be required to validate additional genes and pathways that were identified in the SH-SY5Y screens.

To demonstrate that this platform could be used to identify robust regulators of tau protein, we selected one gene for in vivo validation. In SH-SY5Y cells, knockout of TSC1 and TSC2 increased tau levels. Similar results were obtained in human iNgn2 neurons using two different gRNAs targeting each gene. TSC1 and TSC2 inhibit mTOR[39], and disrupting the TSC complex caused increases in mTOR activation. mTOR's role in regulating tau protein levels has been reported in previous studies. In SH-SY5Y cells and in human AD brains, mTOR localized with phosphorylated tau, and increased levels of mTOR activity were associated with higher levels of soluble and insoluble tau[41–43]. In accordance with previous studies, our study suggests that increases in mTOR activity via disruption of the TSC complex does not increase tau protein levels because of increases in global protein synthesis. TSC complex disruption could impact tau translation via a more selective mechanism, or alternatively could impact tau clearance. Reduction of Tsc1 and Tsc2 in rat neurons did not significantly increase tau aggregation in a 2 week neuronal aggregation assay. In addition, using this same assay, reduction of mTor gene expression did not significantly decrease tau aggregation in rat neurons. This suggests that the magnitude or duration of tau elevation in this model were not sufficient to cause an increase or decrease in accumulated, aggregated tau. In contrast, decreases in TSC2 activity in mice have been shown to increase phosphorylated tau levels[56]. Our data show for the first time that TSC1 negatively regulates tau levels in the brain.

Interestingly, knockout of NCOA6 and FBXW7 reduced tau levels in SH-SY5Y cells, but increased tau levels in neurons. These data suggest that these genes may have a specific role regulating tau protein in post-mitotic neurons. This may be a result of differences in tau levels in SH-SH5Y cells compared to post-mitotic neurons, as tau expression levels increase in Ngn2 neurons during differentiation and peak as neurons mature and extend axons. A second possibility is that the function of NCOA6 and FBXW7 may be different in a neuronal cellular context. NCOA6, is a co-regulator that influences the transcription of multiple genes[57], and FBXW7 is one of the four subunits that form the SKP1-cullin-F-box (SCF) complex[58]. For example, SCFs act as E3 ligases in the ubiquitin proteasome system (UPS). The UPS has been shown to play a unique role in neurons including the clearance of mutant or misfolded proteins, modulating synapse function and axonal degeneration[59]. A third explanation is that Cas9 editing in neurons occurred during neuronal maturation and knocking out genes that impact axonal growth may have an indirect effect on tau levels, however in this case we would expect to see a reduction of tau in neurons.

Additional members of the UPS were identified in this screen, both direct members of the ubiquitination cascade, including UBE4B, UBE2A, and UBE2H, and components of the associated neddylation pathway, including FBXW7, FBXO11, and Cul5 (Table 1; see ref. [34]). Conjugation of ubiquitin to proteins by E1, E2, and E3 ligases, target proteins for degradation by the proteasome[60]. UBE4B, UBE2A, and UBE2H are all E2 ligases. It remains to be determined whether UBE4B, UBE2A, and UBE2H directly regulate tau, or assert their influence indirectly via other pathways. Interestingly, shRNA knockdown of UBE2A and UBE2H also increased levels of α-synuclein in Daoy cells[46], which may indicate a common pathway for E2 ligases regulating proteins with a propensity to aggregate. However, in contrast to its effect on tau, knockdown of UBE4B decreased α-synuclein levels, indicating a specific role of UBE4B in regulating tau protein levels in SHSY-5Y cells.

Neddylation is a process analogous to ubiquitination, where NEDD 8 (a 9KD protein that shares 50% identity with ubiquitin) is conjugated to target proteins by specific ligases. Neddylation of cullin proteins in the SCF promotes ubiquitination of proteins for proteasomal degradation[60,61]. The deneddylation protein SENP8 directly regulates the complexes identified in this screen (SKP1-Cullin1-F-box and Elongin-Cullin5-SOCS-box complexes; see ref. [35]). SENP8 regulates neddylation of cullin proteins and cells deficient of SENP8 cause inefficient degradation of cullin-RING ligase substrates[35]. Disrupting only two of the seven Cullin-RING E3 ubiquitin ligases (CRLs) that are regulated by SENP8 and neddylation, impacted tau levels in this screen. Knockout of components of CRLs 2, 3, and 7 had no effect on tau protein. DCAF7 (part of CRL4A/B) gRNAs were depleted in the unsorted cells at days 21 and 30, consistent with a role in cell proliferation (Supplementary Data 1 and Table 1), therefore its impact on tau levels were not measured using this system. Additional studies are required to understand the specific roles of CRL1 and CRL5 in regulating tau levels, especially given the inverse activity of FBXW7 in neurons. This is the first study that demonstrates a link between neddylation and tau biology, however in AD, neddylation is deregulated and NEDD8 mis-localizes from the nucleus to the cytoplasm[62]. Cells with disrupted neddylation may also have mis-regulated ubiquitination, forcing degradation of tau or α-synuclein proteins via an alternate route such as the lysosome. Indeed, we demonstrate here that knockout of late endosomal proteins, VPS33A and RAB7A, increased tau protein levels in SH-SY5Y cells, and reportedly increased α-synuclein protein levels in Daoy cells[46]. Further evaluation of the role of neddylation and its connection to tau degradation by the proteasome and lysosome may reveal novel pathways for therapeutic intervention.

Several of the initial candidate genes that decreased tau levels in SH-SY5Y cells are chromatin modifying enzymes. Recent work has highlighted the epigenetic modifications in AD as potential therapeutics[63]. For example inhibitors of histone deacetylases or HDACs have been explored to treat cognitive and memory deficits in AD mouse models[64–66], but these inhibitors have yet to successfully translate for use in the clinic[67,68]. Our screen identified and validated NSD1, a histone lysine methyltransferase specific to H3K36 and H4K20[69,70]. NSD1 is expressed in the human brain and has been implicated in prostate cancer, childhood acute myeloid leukemia and Sotos syndrome[71,72]. How NSD1 could be implicated with AD remains to be elucidated, but decreases in methylation have been shown to be involved with AD progression and the formation of neurofibrillary tangles in neurons[63,73]. We find that knockout of NSD1 decreased tau protein levels, suggesting that expression of the *MAPT* gene could be controlled by methylation of H3K36 or H4K20 via NSD1. In addition, NSD1 belongs to the SET2 subfamily of histone methyltransferases which includes NSD2 and 3[72], neither of

which decreased tau levels in our screens, suggesting a unique role of NSD1 in regulating tau expression. Furthermore, CRISPR knockouts of chromatin modifiers, CREBBP, KMT2D and KDM6A decreased tau levels in SH-SY5Y cells and have also been shown to protect human cells from the C9ORF72 dipeptide-repeat-protein toxicity, that causes amyotrophic lateral sclerosis and frontotemporal dementia[74].

Understanding the protein pathways involved in the regulation of endogenous tau, could lead to promising therapies for neurodegenerative diseases. This screen identified and validated proteins and pathways that regulate endogenous tau levels. While targets or proteins that decrease tau levels are more desirable for use as therapeutics, gene knockouts that increase tau levels also advance our understanding of the biology of tau aggregation and neurodegeneration. We have demonstrated for the first time that knock out of TSC1 increased tau levels in the brain identified multiple pathways that regulated tau protein, and provided a valuable resource for the community to further pursue and validate potential targets modulating endogenous tau.

## Methods

**Antibodies**. The following primary antibodies were used to analyze in vitro samples: SP70 (ThermoFisher Scientific #MA5-16404, 1:1000), Cas9 (Diagenode #C15200229, 1:400), M2 FLAG (Sigma # F1804, 1:500), β-actin (Sigma #A5316, 1:1000), Tubb1 (Bio Legend #801201, 1:1000), Map2 (LS-Bio #LS-C61805, 1:1000), Oct4 (Stem Cell #60093, 1:500), Anti-Tau (3-repeat isoform RD3; Millipore #05-803, 1:100), Anti-Tau (4-repeat isoform RD4; Millipore #05-804, 1:100), and Anti-Tau Antibody, clone T49 (Not human; Millipore # MABN827, 1:1000). The following primary antibodies were used to analyze total protein or phosphorylation levels in in vivo studies: Total S6 (Cell Signaling #2317, 1:500), pS6 Ser240/244 (Cell Signaling #5364, 1:2000).

**Generation of SH-SY5Y Cas9 pool and clone 2**. The pNGx_LV_C004 construct derived from pLenti6 (ThermoFisher Scientific #V49610) was used to express Cas9. The construct contains the Cas9 gene from *S. pyogenes* fused to an N-terminal 3×-FLAG tag, and cloned under control of the human cytomegalovirus (CMV) promoter and carries a T2A sequence to a Blasticidin resistance cassette. The construct was packaged, and lenti-virus was used to transduce 200,000 SH-SY5Y parental cells (ATCC CRL-2266) with 500 μl of Cas9 lentivirus in media containing 5 μg/ml of Polybrene (ThermoFisher Scientific #TR1003G) and grown in DMEM F12 (ThermoFisher Scientific #10565-018). Cas9-SH-SY5Y Pool (PC1) was generated from Blasticidin resistant cells (10 μg/ml; ThermoFisher Scientific #A1113903) kept in culture after several passages. The PC1 pooled line was subcloned and multiple single cell clones were derived to form the Cas9-SH-SY5Y clone2 (SC2). Both PC1 and SC2 were analyzed for Cas9 and FLAG expression. Knockout efficiency was assessed by using two short guide RNAs (gRNAs) targeting the *MAPT* gene and a control non-targeting guide. Once guides were delivered via lenti-infection cells were kept under selection using Puromycin (10 μg/ml; ThermoFisher Scientific #A1113803). Editing was allowed for 7, 14, 21, and 30 days. Western blot and ICC was used to determine tau protein levels in edited cells.

**Western analysis of in vitro samples**. Cells were manually lysed in N-Per Buffer (ThermoFisher Scientific #87792) with 1× Halt protease and phosphatase inhibitor (ThermoFisher Scientific #78442) for 10 min at room temperature, followed by a 14,000×g centrifugation at 4 °C to remove cellular debris. Total protein concentrations were measured with the Pierce BCA Assay, following the manufacturer's instructions (ThermoFisher Scientific #23227). Twenty micrograms of protein were prepared with NuPAGE LDS sample buffer (ThermoFisher Scientific #NP0007) and NuPAGE reducing agent (ThermoFisher Scientific #NP0004). Samples were heated at 95 °C for 5 min, and loaded on 4–12% Bis–Tris gels for SDS-PAGE (ThermoFisher #NP0336BOX). Gels were run in 1× MOPS SDS running buffer (ThermoFisher Scientific #NP0001) and transferred to nitrocellulose using a wet transfer system XCellII Blot Module (ThermoFisher Scientific, #EI9051). Membranes were blocked in Odyssey blocking buffer (Licor #927-50000) for 1 h at room temperature. Membranes were incubated with primary antibodies for 16 h at 4 °C diluted in Odyssey blocking buffer. The following day, membranes were incubated 1 h at room temperature with the corresponding species-specific IRDye secondary antibodies diluted in Odyssey blocking buffer and imaged on Licor Odyssey CL.

**Immunocytochemistry (ICC) analysis in vitro**. Cells were fixed with 4% paraformaldehyde (Electron Microscopy Sciences, #15710) for 20 min at room temperature. Cells were then washed with 2× with PBS and permeabilized using Odyssey blocking buffer (Licor #927-50000) with 0.1% Triton (Amresco #0694) for 2 h. Primary antibodies were then added using the same permeabilization buffer

and kept overnight at 4 °C. Cells were then washed 3× with PBS for 5 min. Then secondary antibodies with Hoechst stain (2 μg/ml; ThermoFisher Scientific #33342) were added to the cells and incubated for 2 h at room temperature and away from light. Cells were then washed 3× with PBS and stored at 4 °C or imaged. For analysis and quantification nuclei were defined with Hoechst stain and cells were identified by using CellMask Red (ThermoFisher Scientific # H32712). Image quantification was performed using Cell Profiler[75]. Total tau immunofluorescence intensity was quantified in cells positive with CellMask and segmented from the nuclei signal. For rat neuronal cultures, neurons were washed twice with PBS, and fixed using ice cold methanol (Millipore #67-56-1) for 15 min at −20 °C. Neurons were then washed 3× with PBS and permeabilized using Odyssey blocking buffer with 0.1% Triton for 2 h at room temperature. Primary antibody Tau T49 was then added at 1:1000 in Odyssey blocking buffer with 0.1% Triton and left overnight at 4 °C. Cells were then washed 3× with PBS for 5 min and secondary antibody Alexa Fluor 647 at 1:1000 (ThermoFisher Scientific #A-21236), together with Hoechst stain (2 μg/ml) were added in Odyssey blocking buffer with 0.1% Triton and incubated for 2 h away from light at room temperature. Rat neurons were then washed with 3× with PBS and imaged using in Incell6500 (Cytiva #2-92403-58).

**gRNA library design and construction**. A genome-wide library was designed with five gRNAs against each gene using the Illumina Human BodyMap 2.0 and NCBI CCDS data sets. The genome-wide library contains ~90,000 gRNAs elements covering 18,360 genes. The library was synthesized using chip-based array oligonucleotide synthesis to generate spacer-tracrRNA-encoding fragments[76]. gRNAs were cloned in pool format between BpiI sites of the pNGx-LV-g003 vector derived from pRSI16 lentiviral plasmid by Cellecta Inc. gRNA sequences are driven by the U6 promoter. The UbiC promoter drives the expression of RFP fused via T2A to a Puromycin resistance cassette. Cells transduced with gRNA expressed RFP and can be selected using Puromycin antibiotic. The custom array mini pool library contains 5180 gRNA elements, with ten gRNAs targeting each gene. These were cloned as described for the genome-wide library for the generation of spacer-encoding fragments that were PCR-amplified and then cloned into the BbsI site of pNGx-LV-g003 lentiviral plasmid[23]. gRNA sequences were design to target the most proximal 5′ exons of individual genes. Sequencing of genome-wide library plasmids revealed a normal distribution and passed all quality checks before being packaged into lentivirus.

**Virus production for genome-wide and mini pool custom array libraries**. Libraries were packaged to produce lentivirus using HEK 293T (AF-87-QC67) cells (as described in the ref. [76]). Two hundred and ten million cells were plated on cell bind coated 5-layer stacks (Corning #CLS3311) and left overnight at 37 °C. The following day cells were transfected using 510.3 μl TransIT (Mirus #MIR2300) reagent in 18.4 ml of OPTI-MEM (ThermoFisher Scientific #31985088) that was combined with 75.6 μg of the plasmid pool DNA and 94.5 μg of the Cellecta packaging mix (psPAX2 and pMD2 encoding Gag/Pol and VSV-G respectively; #CPCP-K2A). Seventy-two hours of post-transfection virus was harvested, aliquoted, and frozen at −80 °C. For mini pool, the virus production and reagents were appropriately scaled down and protocols were followed as described for the genome-wide library. Viral titers were measured using RFP by FACS, and were tested on both the cell clones SH-SY5Y Cas9 clonal pool (PC1) and the SH-SY5Y Cas9 single clone (SC2) for genome-wide and mini pool library viruses.

**FACS-based screening approach**. For SH-SY5Y cells (PC1 and SC2 clones), lentiviral multiplicity of infection (MOI) was determined using a ten point dose response ranging from 0 to 400 μl of viral supernatants in 5 μg/ml polybrene. Three days post infection, RFP expressed in infected cells was used to measure infection rate by FACS. Selection by puromycin was optimized by a six point kill curve dose response ranging from 0 to 8 μg of puromycin. 3.7 μg/ml of puromycin was required to achieve >95% death in 72 h. Cell viability was determined using the Titer Glo assay (Promega #G7570) following the manufacturer's protocol.

For genome-wide primary screens, 1.34 million SH-SY5Y Cas9 cells from the clonal pool (PC1) were plated in cell stacks. Twenty-four hours later, cells were infected at 0.6 MOI to allow for one viral particle to infect one cell. Twenty-four hours of post-infection, media was washed out and replaced with media containing 3.7 μg/ml of puromycin. After 72 h under puromycin selection, cells were trypsinized (ThermoFisher Scientific #25200072) and % RFP was measured using FACS to confirm selection of the infected cells. At this point cells were 99% RFP positive and puromycin selection was terminated. Cells were re-plated in new cell stacks and incubated at 37 °C. Cells were incubated for the desired mutagenesis time points (21 and 30 days) and split every 48 h into fresh cell stacks while maintaining 60–70% confluence. No cells were discarded during the editing phase. After mutagenesis time points were met, cells were trypsinized, fixed, and stained for tau using SP70. Cells stained for endogenous tau using SP70 were fixed in 4% paraformaldehyde for 20 min, then washed with PBS, permeabilized in 0.1% Triton X-100 for 10 min, blocked using Odyssey block buffer (LI-COR, #927-70001) for 30 min, and incubated with SP70 primary antibody at 1:100 in 1:1 PBS/Odyssey block buffer for 1 h at room temperature. Then cells were washed with PBS 2–3 times and incubated with Alexa488-tagged secondary antibody (ThermoFisher Scientific #A31565) in the dark for 30 min at room temperature. Finally, cells were

washed 2–3 times with PBS and strained through a 40 μm mesh filter before re-suspending at 30 million cells/ml. Prior to FACS, 1.5 mM EDTA (ThermoFisher Scientific #R1021) was spiked into wash buffers and cells were always kept in ice-cold PBS to avoid cell clumping. As a starting point 2 billion cells were used for tau antibody staining. Because many cells are lost during staining procedure, this yielded 600–800 M cells for FACS sorting. To minimize cell loss from cells sticking to tubes, all tubes were pre-coated with FBS (ThermoFisher Scientific #10438026). FACS was performed on a BD ARIA III. RFP was used to select single cells, and cells were further sorted using Alexa 488 (FITC-A) into the cells with the lowest 25% of tau expression ("low tau" cells) and cells with the highest 25% of tau expression ("high tau" cells) 50 million unsorted cells were also collected as an input sample.

For mini pool library screens SH-SY5Y Cas9 clonal pool (PC1) and single clone (SC2) were plated at 210,000 cells per triple flask. Infection conditions and Puromycin selection was scaled appropriately following the genome-wide screens. One hundred and fifty million cells were collected for every time point and stained with SP70 tau antibody. This yielded 80 million cells for FACS. Two million cells were collected in each of the low tau and high tau FACS sorted cells, and a 2 million cells was collected as an unsorted input control.

**DNA collection and high-throughput sequencing.** DNA from fixed FACS sorted cells was isolated using phenol chloroform extraction and quantified using Quant-iT PicoGreen dsDNA Assay Kit (ThermoFisher Scientific #P11496) following the manufacturer's protocol. Illumina sequencing libraries were generated using PCR amplification with primers specific to the genome integrated lentiviral backbone sequence. For the genome-wide libraries, a total of 24 × 4 μg PCR reactions were performed per sorted sample. For the mini pool custom array libraries, a total of 4 × 2 μg PCR reactions were performed per sorted sample. PCR reactions were performed in a volume of either 100 μl for genome-wide libraries or 50 μl for mini pool custom array libraries containing a final concentration of 0.5 μM of each PCR primer (Integrated DNA Technologies[23]), 0.5 mM dNTPs (ThermoFischer Scientific #N8080261) and 1× Titanium Taq and buffer (Takara Bio #639242). PCR cycling conditions were: 1 × 98 °C for 5 min; 28 × 95 °C for 15 s, 65 °C for 15 s, 72 °C for 30 s, and 1 × 72 °C for 5 min. Resulting Illumina libraries were purified using 1.8× SPRI AMPure XL beads (Beckman Coulter #A63882) following the manufacturer's protocol and quantified using qPCR with primers specific to the Illumina sequences using standard methods. Illumina sequencing libraries were pooled equimolar and sequenced with a HiSeq 2500 instrument (Illumina) using a HiSeq SR Cluster Kit v4 cBot (Illumina, #GD-401-4001) and HiSeq SBS Kit V4 50 cycle kit (Illumina, #FC-401-4002) using 1 × 30 b (gRNA) and 1 × 11 b (sample index) reads. Sequencing was performed following the manufacturer's recommendations (Illumina) using custom sequencing primers (used in the ref. [23]). Libraries were sequenced to a depth such that each gRNA were covered on average by approximately 500–1000 reads.

For genomic DNA isolation of fixed cells 1 million cells were re-suspended in 250 μl TNES (10 mM Tris-Cl pH 8.0, 100 mM NaCl, 1 mM EDTA, 1% SDS) and incubated at 65 °C overnight. Following incubation, 5 μl 10 mg/ml RNase A was added and incubated for 30 min at 37 °C. To this mixture, 5 μl 10 mg/ml Proteinase K was added and incubated for 1 h at 45 °C. To this mixture, 250 μl Phenol:Chloroform:Isoamyl alcohol (PCIA, 25:24:1) was added. Tubes were vortexed and spun in a microcentrifuge at full speed for 2 min. The aqueous phase was transferred to 250 μl PCIA and vortexed and spun again as before. The aqueous phase was then transferred to 225 μl Chloroform and vortexed and spun again. Following this, the aqueous phase (about 200 μl) was transferred to 20 μl 3 M NaAcO pH 5.2. To this, 500 μl Ethanol was added and incubated on ice for 1 h. The cold mixture was then spun in a microcentrifuge at full speed to pellet the DNA. The DNA pellet was washed in 500 μl 70% Ethanol and dried fully. Finally, the dried pellet was re-suspended in 25 μl water and the DNA was quantified using a Nanodrop.

**STRING enrichment analysis.** STRING was used to determine statistical enrichment of pathways in gene sets generated from the screens[31]. Two separate gene sets for this analysis were compiled from gene knockouts that downregulated tau and gene knockouts, that upregulated tau from at least two of the mini pool custom array validation screens with an RSA score cut-off of ≤−5. STRING network edge parameters were set at high confidence (0.700) and are represented by the line thickness between nodes. The gene sets were analyzed using multiple sources including Reactome, KEGG, Gene Ontology, and published literature. Networks were determined based on low false discovery rate (<1.2%) and confidence between nodes ranging from high to highest (0.700-0.900) (Supplementary Data 4 and 5).

**Generation of single gRNA clones and viral packaging.** The top two performing single gRNA sequences were selected from the primary screen (30 day mutagenesis) for additional neuronal validation. Individual sequences were cloned into the BbsI site of pNGx-LV-g003 vector using golden gate cloning. Plasmids were sequenced after cloning to verify gRNA sequence was successfully cloned. RFP and puromycin cassettes in the vector were used to confirm infection and selection after neuronal infection. DNA was transformed into One Shot TOP10 competent E. coli

cells (ThermoFisher Scientific #C404010) and plated into Ampicillin plates. The next day 1–2 colonies were selected and grown on 5 ml of LB Broth (ThermoFisher Scientific #10855001) with 50 μg/ml of Ampicillin. DNA was extracted by mini preps using the QIAcube (Qiagen #990395). Packaging was done in 96-well plate setting, briefly HEK293 cells (AF-87-QC67) were plated in collagen coated 96-well plate (BD Biosciences #356407) at 33,000 cells per well in DMEM (ThermoFisher Scientific #11965) supplemented with 10% FBS (ThermoFisher Scientific #16000044) and 1% NEAA (ThermoFisher Scientific #11140050) with no pen/strep and incubated at 37 °C overnight. The next day cells were transfected with 0.6 μl TransIT (Mirus #MIR2700) reagent diluted in 9.4 μl of OPTI-MEM (Thermo Scientific #31985088) combined with 100.5 ng of plasmid DNA with gRNA sequence and 110 ng of the Cellecta packaging mix (psPAX2 and pMD2 encoding Gag/Pol and VSV-G respectively; #CPCP-K2A) per well and left at 37 °C overnight. The following day, after transfection, media was replaced with 200 μl per well of DMEM supplemented with 10% FBS and 1 g/100 ml of BSA (ThermoFisher Scientific #AM2616) with no pen/strep. Forty-eight hours after media change, supernatant for every well was harvested and transferred to another 96-well plate and stored at −80 °C until ready to use.

**PDL laminin coated plates for iNgn2 neurons.** A PBS mix of Laminin diluted 1:300 (Sigma #L2020) was dispensed at 100 μl per well of PDL coated 96-well plates (Corning Cat#354640) and incubated overnight at 37 °C. Plates were then washed six times with PBS, by not allowing the plate to dry during washes. PBS was left on the wells after the final wash and removed when cells were added.

**Maintenance and preparation of iNgn2 neurons.** Human ESCs[36] were maintained in mTeSR media (Stem Cell Technologies #85850) on 1× Matrigel hESC-Qualified Matrix at 1:100 (Corning #354277) until ready for differentiation. For induction mTeSR media was changed to DMEM/F12 + Glutamax (ThermoFisher Scientific #12491) with 1% Pen/Strep (ThermoFisher Scientific #15240-062), 1% NEAA, 0.5% N2 Supplement (ThermoFisher Scientific #17502-048), Doxycycline (2 μg/ml; ClonTech #631311), BDNF (10 ng/ml; PeproTech #450-02) NT3 (10 ng/ml; PeproTech #450-03). After 72 h cells were incubated with Accutase (ThermoFisher Scientific #A1110501) for 7–10 min at 37 °C to dissociate and were then plated in 96-well plates previously coated with PDL laminin at 75,000 cells per well using DMEM/F12 + Glutamax with 1% Pen/Strep, 1% NEAA, 0.5% N2 Supplement, Doxycycline, BDNF, NT3, and B27 minus Vitamin A (1:50; ThermoFisher Scientific #12587010). Forty-eight hours after re-plating cells were infected with specific gRNA lentivirus. Puromycin at 1 μg/ml was then added to the cells 72 h later. Cells were maintained at 37 °C with half media changes every 72 h. Fourteen days after lentiviral infection, cells were prepared for Human Tau AlphaLISA. Cells were lyzed in N-Per Buffer (ThermoFisher Scientific # 87792) with 1× Halt protease and phosphatase inhibitor (ThermoFisher Scientific #78442) for 10 min at room temperature, followed by a 14,000 × g centrifugation at 4 °C to remove cellular debris. Total protein concentrations were measured with the Pierce BCA Assay, following manufacturer's instructions (ThermoFisher Scientific #23227).

**Human tau AlphaLISA.** HT7b/BT2 sandwich AlphaLISA was used to measure tau protein levels in iNgn2 neurons following treatment with gRNAs. Briefly, human recombinant Tau 441 was used for standard curve (Perkin Elmer #AL271S) prepared in a 12-point dilution curve (2-fold dilutions) diluted in 1× Immunoassay buffer (Perkin Elmer #AL00F). Biotinylated mouse anti-Tau monoclonal antibody HT7 (ThermoFisher Scientific #MN100B: 100 μg/ml) and a mouse anti-Tau monoclonal antibody BT2 (ThermoFisher Scientific #MN1010) was conjugated to Perkin Elmer AlphaLISA acceptor beads (Perkin Elmer, # 6772002), to a final 5 mg beads/ml concentration. Both tau antibodies were prepared in 1× Immunoassay buffer by diluting each antibody 1/100 in buffer. Ten microliter of recombinant tau or diluted sample was incubated with 10 μl per well tau antibody mixture in 384-well plate, shaking overnight at 4 °C. Following incubation, 10 μl per well of streptavidin conjugated donor bead (Perkin Elmer, #6760002S) diluted 1/25, final concentration 200 μg/ml, in 1× Immunoassay buffer was added to plate for 1 h at room temperature. Following incubation, AlphaLISA counts were measured on EnVision plate reader (Perkin Elmer). Tau levels were calculated according to a nonlinear regression using the 4-paramter logistic equation (sigmoidal dose-response curve with variable slope) and a $1/Y^2$ data weighting. The tau protein levels are measured in pg/ml, adjusted for sample dilution factor, and then normalized to total protein concentration. Results were represented as fold change of tau protein relative to control (Non-targeting) gRNAs.

**Animal model.** For in vivo validation of a neuronal model of mTOR hyperactivity, a $Tsc1^{fl/fl}$; CamKIIα-Cre (Tsc1 cKO hereafter) mouse model was used. The $Tsc1^{fl/fl}$ (Brendan Manning Lab, Harvard University) mouse line was crossed to the CamKIIα-Cre line[40]. As previously described, these mice exhibit increased mTOR signaling and mortality before P60. All in vivo research was reviewed and approved by the Novartis Institutes of Biomedical Research Institutional Animal Care and Use Committee in accordance with applicable local, state, and federal regulations.

**Western analysis of mouse brain tissue.** Hemi-forebrains were rapidly dissected and frozen in liquid nitrogen. Tissue samples were homogenized in T-Per Buffer

(ThermoFisher Scientific #78510) including protease inhibitors (Roche #490683700) and phosphatase inhibitors (Roche #4693159001) using the Precellys24 bead homogenizer. Following homogenization, a 14,000×$g$ centrifugation step was used to remove cellular debris. Protein concentrations in the resulting supernatant were assayed by modified Lowry method using manufacturer's recommendations (BioRad DC kit II #5000112. Samples were prepared with NuPAGE LDS sample buffer (ThermoFisher Scientific #NP0007) and NuPAGE reducing agent (ThermoFisher Scientific #NP0004) and denatured at 95 °C for 5 min Phosphorylation of mTOR substrates was determined in 56 µg protein samples subjected to 4–12% SDS-PAGE (Bis–Tris, Bio-Rad #3450124), followed by a transfer to nitrocellulose membranes (IBlot2 System #IB23001). Membranes were blocked in Licor blocking buffer (Licor #927-40000) at room temperature for 4 h and incubated in primary antibody overnight at 4 °C. Membranes were incubated with species-specific fluorophore-conjugated secondary (Licor anti-rabbit #926-68021 and anti-mouse #926-32210) antibodies for 1 h at room temperature and antibody signal was detected using the Licor Odyssey.

**Mouse total tau ELISA.** Mouse total tau levels where determined with the Meso Scale Discovery (MSD) Mouse total tau assay (MesoScale Discovery #K151DSD). The assay was used according to the manufacturer's instructions. Briefly, 150 µl/well of blocker A was added to each well of a MSD plate and incubated at room temperature with shaking for 1 h. During incubation, the calibration curve and samples were diluted in blocker A as per manufacturer's instructions. After 1-h incubation, plates were washed three times with 150 µl with 1× MSD Wash Buffer (Meso Scale Discovery # R61AA-1). Twenty-five microliter of previously prepared calibrator and samples were added per well and incubated at room temperature for 1 h with shaking. After sample incubation, plates were washed three times with 150 µl/well of 1× MSD wash buffer. Twenty-five microliter of 1× antibody detection solution, prepared per manufacturer's instructions was added to each well and incubated at room temperature for 1 h with shaking. After secondary incubation, plates were washed three times with 150 µl/well of 1× MSD wash buffer. Finally, 150 µl/well of 1× Read Buffer T was added to each well and analyzed immediately on the SECTOR Imager 6000.

**In vitro rat inclusion assay.** Rat cortices were dissociated using a Papain Dissociation System (Worthington Biochemical Corporation, cat#LK003150) following manufacturer's instructions. Briefly cortices were incubated with 1 ml of papain solution and 100 µl of DNase solution for 10 min at 37 °C. Tissue settled and solutions were removed and set aside, then cortices were re-incubated using 1.5 ml of papain solution and 150 µl of DNase solution for 10 min at 37 °C. Solutions from both time points were combined and tissue was then centrifuged at 1200 rpm for 4 min. Supernatant was removed and cells were re-suspended in 8 ml of Ovomucoid inhibitor and centrifuged at 1200 rpm for 4 min. Supernatant was removed and cells were re-suspended in Neurobasal Plus Media (ThermoFisher Scientific #A3582901), supplemented with 1% Glutamax (ThermoFisher Scientific #12491), 1% Pen/Strep (ThermoFisher Scientific #15240-062) and 2% B27 Plus (ThermoFisher Scientific # A3582801). Cells were counted and plated in a PDL coated 96-well plate (Corning #354640) at 40,000 cells in 150 µl of media per well and incubated at 37 °C. Five days after plating, half the media was changed per well and 1.5 µM of Accell siRNA was introduced per well. Accell siRNAs used: Non-Target (Horizon Discovery #D-001910-10-05), MAPT (Horizon Discovery #E-089500-00-0005), Tsc1 (Horizon Discovery #E-096662-00-0005), Tsc2 (Horizon Discovery #E-091541-00-0005) and mTor (Horizon Discovery #E-090701-00-0005). Two days after siRNA addition, 1.5 µl of human AD tau seeds was introduced to each of the wells that were incubated with siRNA. Seven days of post-seed incubation a half media change was performed. Methanol fixation of tissue cells was then performed 7 days after second half media change (21 days of post cell plating). ICC using tau T49 antibody was done and cells were imaged using Incell 6500.

**Purification of tau seed from AD brains.** AD tau seeds were extracted from human brains by following the purification of tau seed protocol[44]. Briefly, a high salt lysis buffer was prepared using 10 mM Tris-HCL pH 7.4, 0.8 M NaCl, 1 mM EDTA, 2 mM Dithiothreitol (DTT), 10% Sucrose, 0.1% Sarkosyl, Roche Complete tablets (Cat# 04693159001) and Roche PhosSTOP tablets (Cat# 04906837001). Buffer was prepared based on tissue weight multiplied by 9 µl of lysis buffer. Tissue was homogenized in lysis buffer using a Precellys homogenizer. Tissue was centrifuged at 10,000×$g$ for 10 min at 4 °C. Supernatants were collected and pellet resuspended in 2/3 volume lysis buffer before re-centrifuging at 10,000×$g$ for 10 min at 4 °C. Supernatants were pooled together and 1% sarkozyl added before incubating for 1 h at 4 °C. The samples were then centrifuged at 300,000×$g$ (50,000 rpm) for 1 h at 4 °C using a VCM rotor, (Beckman 50.2Ti). The pellet was washed once with PBS and resuspended in PBS. This was sonicated with 20–60 short pulses (0.5 s/pulse), and finally centrifuged at 10,000×$g$ for 30 min at 4 °C. The supernatant was the final AD tau seed (sarkozyl insoluble fraction) used in the assay.

**Taqman qPCR for knockdown expression.** Lysates and multiplex RT-PCR were collected and performed using the FastLane Cell Multiplex Kits (Qiagen #216513) and following manufactures instructions. The following Taqman probes used were: MAPT (ThermoFisher Scientific #Rn00691532), Tsc1 (ThermoFisher Scientific #Rn00573107), Tsc2 (ThermoFisher Scientific #Rn00562086), mTor (ThermoFisher Scientific #Rn00693900) and GAPDH (ThermoFisher Scientific #Rn01775763). Percent knockdown for each gene was calculated by determining quantification cycle (Cq) for both the target gene and GAPDH in cells treated with either siRNA targeting gene of interest (MAPT, Tsc1, Tsc2, or mTor) or for non-targeting control. Delta Cq (ΔCq) was determined as the difference between Cq of the target gene minus the Cq of GAPDH. ΔCq was then exponentially transformed using $2^{-\Delta Cq}$ to determine ΔCq expression for each targeted gene in siRNA treated samples and non-target controls. Technical replicates were then averaged per gene and normalized to ΔCq expression of the target gene in a non-target control treated cells (ΔΔCq). Percent knockdown was then calculated for each targeted gene using $(1 - \Delta\Delta Cq) \times 100$.

**Statistics and reproducibility.** For genome-wide primary screens two technical replicates were performed for each editing time point (21 and 30 days) using the genome-wide library. For mini pool library screens six technical replicates per cell clone (PC1 and SC2) per time point (14, 21, and 29 days) were performed using custom array mini pool libraries.

For statistical analysis of screening data the raw gRNAs sequencing reads (Supplementary Data 7 and Supplementary Data 8) were aligned to the appropriate libraries to generate gRNA-level counts. The counts were used to calculate a log2 (fold change) between the low tau cells samples and the high tau cells samples using DESeq2[77]. Robust $z$-scores were then calculated for each gene using the fold changes for all the gRNAs for that gene. The gRNAs for each gene were ordered according to fold changes, and the gRNA that was closest to the Q1 quartile (second ranked gRNA) or the Q3 quartile (fourth ranked gRNA) were identified. The quartiles Q1 or Q3 were plotted versus Redundant siRNA activity (RSA) scores. The RSA analysis calculates a –log10 $p$-value based on the ranked performance distribution of all the gRNAs targeting a specific gene[28]. Lower RSA scores for a specific gene (higher significance) means there were multiple active gRNAs showing the same phenotype. Activity of each gRNA was determined by calculating the fold change (FC) of sequence reads counts representation in the high tau versus low tau FACS sorted cells. The FC was log2-tranformed (log2(FC)) and further transformed with the robust $z$-score calculation (each gRNA FC was subtracted by the median FC for all tested gRNAs then divided by the median absolute deviation of all gRNAs FC). The fold change of the unsorted cell population compared to the input library was also computed to assess effects on proliferation.

For neuronal Ngn2 validation experiments, seven biological replicates were performed. Total tau levels in each well lysate were first normalized to total protein per well. Each replicate plate was normalized to its own non-target control wells to account for inter-plate variability. For each plate, total tau per well was divided by the mean of all the non-target control wells. The results of the normalizations for all seven individual replicates were then pooled and the median of the normalized total tau was calculated per gRNA. Z-scores for each gRNA were then calculated using the normalized medians by calculating the mean and standard deviation of the whole set. To determine if a candidate gene validated in our neuronal model both gRNAs for each candidate gene needed a $z$-score of at least one standard deviation from the mean.

For western blot analysis of mouse brains, to quantify mTOR pathway activity, pS6 or pAKT band intensities were normalized to total protein levels. Statistical analysis for phosphorylation changes were analyzed by one factor ANOVA, followed by Tukey's post-hoc comparisons. For Mouse tau ELISA analysis, total tau levels were calculated according to a nonlinear regression using the 4-paramter logistic equation (sigmoidal dose-response curve with variable slope) and a $1/Y^2$ data weighting. The total tau levels measured in ng/ml were adjusted for assay dilution factor and then normalized to total protein concentration, recorded as ng total tau/mg protein. Statistical analysis for changes in tau levels were analyzed by one factor ANOVA, followed by Tukey's post-hoc comparisons.

Images were analyzed for inclusion counts using CellProfiler[75]. Inclusions were identified using the Tau T49 intensity and size and nuclear counts were identified using Hoeschst stain. Nine fields were taken per well. Each field was then analyzed to determine inclusion (tau aggregate) count and nuclear counts. All fields per well were then pooled to determine the total number of inclusions and nucleus quantified per well. Background of Tau T49 stain was filtered out by comparing wells with no seed controls, and determining how many inclusions are picked up by cell profiler in these wells. The total number of inclusions per well were then normalized to the number of inclusions in non-target control wells to determine fold change relative to non-target controls. This experiment was done in four biological replicates and one-way ANOVA test was performed to determine significance.

**Reporting summary.** Further information on research design is available in the Nature Research Reporting Summary linked to this article.

## Data availability

Correspondence and requests for materials related to this study should be sent to fiona.elwood@novartis.com. All the source data supporting the findings of this study are available within the paper and in Supplementary Data 1–9. The original blot image for

Fig. 1c is included in Supplementary Fig. 5a. The FACS gating strategy is shown in Supplementary Fig. 6. Any other relevant data are available from the authors on reasonable request.

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

## Acknowledgements

The authors are grateful to Brendan Manning for sharing the Tsc1fl/fl mice, Dimple Doshi for experimental support and Frederic Sigoillot for collection/packaging of source data.

## Author contributions

C.G.S., C.M.A., and F.E. wrote the manuscript. C.G.S. performed neuronal validation. C.M.A., A.G., L.C., and K.C. performed in vivo experiments. M.V., C.S., and C.D. characterized cell lines used in the screens. M.V., C.S., and N.R.C. performed the primary screens. S.P. performed FACS for screens. A.L., C.R., and G.M.A. performed next generation sequencing and enrichments analysis. J.A., J.R.H., and S.A. constructed CRISPR libraries. S.J.L., G.R.H., M.P., and R.D. conceptualized screening experiments.

## Competing interests

All authors are/were employees of Novartis Institutes for BioMedical Research at the time this work was performed.
