## [Peer Review File · Communications Biology]

Reviewers' Comments:

Reviewer #1:

Remarks to the Author:

This study by Sanchez et al is a very comprehensive CRISPR screen for modifiers of tau levels. The authors have performed a CRISPR screen in SH-SY5Y cells followed by validation in Ngn-2 induced human neurons and a transgenic animal. Overall, the screen was performed at a very high level of quality and the data are clearly presented. As such the study represents a very valuable resource for the research community.

The authors need to include a paragraph explaining some of the limitations of the study in particular in regard to the maturity of the neuronal systems used as none is likely to express tau isoforms in the proportions found in the human brain.

Why was the screen performed at day 21 and day 30? Figure 1 shows sufficient editing at least for tau at day 14. Because of this time latency, it is likely that most or all of the hits are influencing tau levels by an indirect effect and this issue needs to be discussed.

It would be useful for the reader to discuss how TSC1 knockout causes increased tau levels. Is it through effects on translation? Can this be tested?

In Figure 1 there seems to be a discrepancy between the immunoblotting in panel c showing almost complete depletion of tau and immunostaining in panel d suggesting a reduction by about 50%. Please explain.

In Figure 3, please include images of the differentiated neurons and their basic characterisation with neuronal markers either as a panel or supplement. As stated above the authors need to discuss the limitation of this model in terms of maturity and tau isoform expression.

Figure 4: please explain what the dots represent: technical replicates from the same animal or different animals? Each data-point should be one animal

Reviewer #2:

Remarks to the Author:

In the manuscript titled "Genome wide CRISPR screen identifies protein pathways modulating tau protein levels in neurons", the authors reported the identification of several pathways affecting expression level of Tau in neuronal cells by CRISPR-mediated whole genome screening. As a pathogenic protein in AD, it is important to understand the genetic regulators of pathogenic Tau protein levels. The CRISPR based gene edition in this study is well designed and efficient. However, the poor robustness of screening model, the low disease relevance and the lack of mechanistic exploration make the current report unsuitable to be accepted.

Major comments:

1. For the Genome wide CRISPR screen, the screen model itself is a very important part of the study. The staining of endogenous tau in SH-SY5Y cells is a very simple and low efficiency model. Moreover, Tau plays an important role in neuronal function and only the hyper-phosphorylated tau is linked with AD. It is very difficult to appreciate the strategy to regulate total tau, but not hyper-phosphorylated tau.
2. The study is a simple report of findings without any mechanistic exploration. Identification of candidate genes/pathway is just the beginning. Authors need to reveal the underlying mechanism by which the genes regulate Tau protein level. Furthermore, the potential effects of these genes on the Tau mice AD models need to be examined to establish the disease relevance.

Reviewer #3:

Remarks to the Author:

The manuscript by Sanchez et al. from the Novartis Institutes for BioMedical Research, Cambridge MA, reports a genome-wide pooled CRISPR knockout screen to identify novel genes and pathway modulators of endogenous tau protein levels. A primary screen was carried out in the SH-SY5Y neuroblastoma cell line, followed by validation in Ngn2-induced human cortical neurons.

The primary screen unveiled 266 genes whose knockout decreased and 215 genes whose knockout increased tau protein levels. Approximately 30% of the genes, which were identified to lower or raise tau levels in the primary screen, were confirmed in a secondary mini-pool screen undertaken in Ngn2. Most of these genes are understood to play roles in protein ubiquitination, chromatin modeling, neddylation, late endosomal biology, RNA biology, TGF-beta/BMP and mTOR signaling. Tuberous sclerosis protein 1 (TSC1), a component of the mTOR pathway, passed subsequent in vivo validation as a gene whose knockout increases tau levels.

The study is timely in light of ongoing efforts to devise methods for the suppression of key neurodegenerative disease proteins. Although conceptually sound and based on leading-edge technology, this reviewer was disappointed to find that the experimental paradigm for executing the primary screen centered on the SH-SY5Y cell model, an inferior model for studying tau biology. The limits of this model for drawing conclusions about neurons is apparent from the results, which revealed that only "3 of the top 20 genes reducing tau levels and 2 of the top 23 genes increasing tau levels validated in human neurons." That said, the study is well conducted, critical controls were included at key steps along the way, and the manuscript is a pleasure to read. The exception to the latter statement is a convoluted paragraph on the ranking analyses that the authors are advised to subject to revision.

Minor Concerns:

- Although both a pooled SH-SY5Y model as well as a clone from this cell line was available, this reviewer found it in places less than clear, which of these two paradigms was used for which of the analyses.
- A brief characterization of tau isoforms and the occupancy of key tau phosphoepitopes in the SH-SY5Y, Ngn2 paradigms, and in human brains should be included, and the significance of differences for the interpretation of results should be discussed.
- Line 35 "OR" transition isn't clear – Could be rephrased to OR [Line 38 - "secondary" where the tau lesion is downstream of a causative insult such as AB amyloid plaques in Alzheimer's disease (AD)]. Another option is adding the word "when" following OR.
- Line 35: inaccurate wording using "only"- tau is not the ONLY aggregated protein found in brains of patients with PSP, CBD, PiD, and FTDP-17. Consider replacing with "predominant".
- Lines 135, 159, 207, 208, and other instances: Font error (temperature unit).
- Lines 573, 576, and other instances: brackets left open
- Line 594: If only the top 20 of 93 genes reducing and top 23 of 62 genes increasing tau levels were validated in human neurons, what was done with the remaining genes, and what did their bioinformatic evaluation reveal?
- A brief paragraph reminding the reader of the link of TSC1 to the mTOR pathway would be helpful.

Taken together, this reviewer remains enthusiastic about this study and is supportive of its publication once it has been further strengthened by addressing the comments raised here.

Review 1

Reviewer Comments: The authors need to include a paragraph explaining some of the limitations of the study in particular in regard to the maturity of the neuronal systems used as none is likely to express tau isoforms in the proportions found in the human brain.

Response: The paragraph below discussing the limitations and maturity of our neuronal system used to validate screen hits has been added to the discussion.

“The low validation rate in neurons may be due to the intrinsic differences between the SHSY5Y cell line and Ngn2 human neurons, or it may reflect the fact that editing in human neurons occurred during differentiation of the neurons. While Ngn2 neurons offered the ability to test cell type specific phenotypes in a pure population of excitatory glutamatergic neurons without having to account for the possibility of a diluted phenotype from mixed neuronal cultures^{42, 54}, the system has its limitations. At 14 days post differentiation, the iNgn2 neurons did express the 4R tau isoform and levels of total tau protein were not comparable to human brain levels. Previous studies have attempted to address this Ngn2 limitation by culturing neurons for longer. This increased the levels of 4R tau, but the 3R:4R Tau expression ratio was not comparable to expression ratios in human brains⁵⁵. Using an induced pluripotent stem cells (iPSC)-neuronal system from patients with tau mutations, differentiation for 150 to 365 days was required to see expression of multiple tau isoforms, yet these still did not compare to the levels seen in the human brains^{56, 57, 58}. While a more robust human neuronal system with expression of both 3R and 4R tau isoforms would benefit studies looking at tau aggregation, the scope of this study is only looking at regulators of tau protein levels. For these purposes, the current neuronal system allowed for a rapid additional hit validation in a human post-mitotic neuronal system that expressed endogenous tau levels and triage of hits for validation in a more mature in-vivo neuronal system. Additional approaches such as CRISPR-i or pharmacological inhibition in differentiated human neurons⁵⁵, or in vivo experiments may be required to validate additional genes and pathways that were identified in the SH-SY5Y screens.”

Reviewer Comments: Why was the screen performed at day 21 and day 30? Figure 1 shows sufficient editing at least for tau at day 14. Because of this time latency, it is likely that most or all of the hits are influencing tau levels by an indirect effect and this issue needs to be discussed.

Response: The initial time points of 21 and 30 days for the primary screens were selected because of the previously reported half life of tau being 6-23 days (Sato et al., 2018) These time points would allow for ample time for the protein being knockdown to have an effect on tau. Validation mini pool library screens were performed with 3 time points, 14, 21 and 29 days to make sure hits from the primary screens could be validated at all 3 time points. The text has been revised to make these points more clear.

“After infection, cells with stable gRNA integration were selected, and edited for 21 or 30 days. These time points were selected because previous studies reported the half-life of tau protein between 6-23 days³⁵, therefore, we wanted to allow ample time for the protein being knocked down to have an effect on tau protein levels.”

“Since both cell lines showed nearly complete tau protein knockdown after 14 days of editing using MAPT gRNAs (Figure 1c, Supplementary Figure 1a), and to address the possibility that influences on tau protein levels could be an indirect effect, an additional 14 day editing time point was added to the validation screens. Therefore cells were edited for 14, 21 and 29 days (Supplementary Figure 3a).”

Reviewer Comments: It would be useful for the reader to discuss how TSC1 knockout causes increased tau levels. Is it through effects on translation? Can this be tested?

Response: In our neuronal and in-vivo mouse experiments when we knocked out components of the TSC complex, we consistently observed increases of tau protein levels, but did not observe an increase in over all protein levels. A paragraph discussing this has been added to the text.

“In accordance with previous studies, our study suggests that increases in mTOR activity via disruption of the TSC complex does not increase tau protein levels because of increases in global protein synthesis. TSC complex disruption could impact tau translation via a more selective mechanism, or alternatively could impact tau clearance. Reduction of Tsc1 and Tsc2 in rat neurons did not significantly increase tau aggregation in a 2 week neuronal aggregation assay. In addition, using this same assay, reduction of mTor gene expression did not significantly decrease tau aggregation in rat neurons. This suggests that the magnitude or duration of tau elevation in this model were not sufficient to cause an increase or decrease in accumulated, aggregated tau. In contrast, decreases in TSC2 activity in mice have been shown to increase phosphorylated tau levels⁵⁹. Our data show for the first time that TSC1 negatively regulates tau levels in the brain”

Reviewer Comments: In Figure 1 there seems to be a discrepancy between the immunoblotting in panel c showing almost complete depletion of tau and immunostaining in panel d suggesting a reduction by about 50%. Please explain.

Response: The differences between the western blots (WB) in Figure 1c and Supplementary Figure 1a, and the immunocytochemistry (ICC) panels in Figure 1d is of 7 days. ICC was done 7 days post infection while WBs were done at 14, 21 and 30 days post infection. It makes sense that this would be the case as at 7 days post infection you would still have some perdurance of the tau protein. This has been re-arranged in the text to fall in line with the figure panels.

“To measure Cas9 editing efficiency and subsequent tau reduction, western blot analysis was performed for tau protein levels from lysates of PC1 and SC2 cells. A significant reduction of tau protein in both PC1 and SC2 cells could be seen 14 days post infection; and this reduction was maintained at 21 and 30 days post gRNA infection (Figure 1c, Supplementary Figure 1a). On average, 85% or 75% reduction of tau was maintained over the time period examined with gRNA MAPT-1 and MAPT-2 respectively (data not shown). In addition, immunocytochemistry (ICC) was performed using a total tau antibody, SP70. 50% reduction of tau protein was observed as early as 7 days post gRNA infection (Figure 1d).”

Reviewer Comments: In Figure 3, please include images of the differentiated neurons and their basic characterization with neuronal markers either as a panel or supplement. As stated above the authors need to discuss the limitation of this model in terms of maturity and tau isoform expression.

Response: ICC for common neuronal markers, Tubb3 and MAP2; and for stem cell marker, Oct4, have been added to Supplementary Figure 4a. In addition, ICC for Tau (Sp70) and Cas9 were added to Figure 3a. The text was revised accordingly, incorporating the newly added figure panels.

Reviewer Comments: Figure 4 please explain what the dots represent: technical replicates from the same animal or different animals? Each data-point should be one animal

Response: Sentence has been added to the legend on Figure 4, saying that each data point represents one animal,

“Each data point represents an individual animal.”

Review 2

Reviewer Comments: For the Genome wide CRISPR screen, the screen model itself is a very important part of the study. The staining of endogenous tau in SH-SY5Y cells is a very simple and low efficiency model. Moreover, Tau plays an important role in neuronal function and only the hyper-phosphorylated tau is linked with AD. It is very difficult to appreciate the strategy to regulate total tau, but not hyper-phosphorylated tau.

Response: A paragraph discussing the importance of finding regulators of tau protein levels and not specific hyper-phosphorylated epitopes has been added to the introduction of the manuscript.

“Tau pathology can cause neurodegeneration independent of A β in patients suffering from primary tauopathies. MAPT mutations are present in approximately 5% patients with FTDP-17^{6, 7, 8, 9}. While aggregated, extensively post-translationally modified species of tau are present in diseased brains, the exact forms that are the most toxic remain a matter of debate. Interestingly, reducing or ablating endogenous tau levels in mice is well tolerated^{10, 11, 12, 13}, and therefore several therapeutic approaches that aim to reduce total tau levels have been pursued¹⁴. These include targeting the mRNA with an antisense oligonucleotide (ASO)¹⁴ and using zinc finger proteins and gene therapy to target tau expression levels¹⁵. Other approaches have attempted to target extracellular tau using passive or active immunotherapy (reviewed in¹⁶), or drive degradation of aggregated tau by activation of the proteasome pathway^{17, 18, 19}. These novel modalities are at various phases of drug development from early preclinical testing to phase II clinical trials¹⁶. Because of the high levels of tau molecular diversity in AD^{20, 21}, one major advantage of tau lowering approaches, is that they can potentially reduce all species of tau proteins, including post translationally modified tau species that cause pathology. However, one disadvantage of the current tau therapies is that they rely predominantly on large molecules, including antibodies and ASOs that have poor access to the brain. Therefore, there is great interest in identifying targets that regulate the expression or degradation of tau and can be modulated with small brain penetrant molecules to reduce pathogenic tau in patients.”

Reviewer Comments: The study is a simple report of findings without any mechanistic exploration. Identification of candidate genes/pathway is just the beginning. Authors need to reveal the underlying mechanism by which the genes regulate Tau protein level. Furthermore, the potential effects of these genes on the Tau mice AD models need to be examined to establish the disease relevance.

Response: This manuscript is meant to be reported as a resource to the community. Here we used a validated screening strategy to find regulators of total tau protein. For this purpose we have validated a screen hit in a mouse system and shown in this system that disrupting TSC1 indeed also increases tau protein levels. To address some of the concerns of the reviewer we tested if the TSC complex and mTOR knockdowns affect tau aggregation levels in rat neurons using a well reported tau aggregation assay. Using this assay we observe no differences in tau aggregation levels in rat neurons that are TSC1, TSC2 or mTOR deficient, the data of this experiment has been added to Supplementary Figure 4g-i.

Review 3

Reviewer Comments: Although both a pooled SH-SY5Y model as well as a clone from this cell line was available, this reviewer found it in places less than clear, which of these two paradigms was used for which of the analyses.

Response: The text has been edited to address this concern. For every experiment on the paper that uses either cell line its is stated properly as either the pooled PC1 cell line or the single clone SC2 cell line. Examples of how the text has been edited in various parts of the manuscript can be seen below. The figures and figure legends have been edited as well.

“Two SH-SY5Y neuroblastoma cell lines that endogenously express total 3R tau, and stably expressed FLAG-Cas9 were used (Figure 1b, c, Supplementary Figure 1a, e). The first, being a pooled cell line, hereafter referred as PC1; and the second being a single clone expanded from the PC1, hereafter referred as SC2.”

“Validation screens were performed on two independent cell lines, the pooled PC1 cell line (used in primary screens), and a cell line derived from a single clone of PC1, the SC2 cell line. These two cell lines were used to make sure results could be recapitulated in different cellular contexts.”

Reviewer Comments: A brief characterization of tau isoforms and the occupancy of key tau phosphoepitopes in the SH-SY5Y, Ngn2 paradigms, and in human brains should be included, and the significance of differences for the interpretation of results should be discussed.

Response: Characterization of the tau isoforms using western blots for each of the SHSY-5y cell lines used in the primary screens and the NGN2 neurons used for validation have been added to Supplementary Figure 1e and Supplementary Figure 4b, respectively. Our screen was looking for regulators of total tau protein levels regardless of the phosphorylation status of the protein. We believe that by finding regulators that affect the total protein you can also affect specific tau phosphoepitopes. A sentence making this point has been added to the introduction of the manuscript

“Because of the high levels of tau molecular diversity in AD^{20,21}, one major advantage of tau lowering approaches, is that they can potentially reduce all species of tau proteins, including post translationally modified tau species that cause pathology.”

In addition, a paragraph discussing the limitations of our neuronal model used to validate screening candidates has been added to the discussion.

“The low validation rate in neurons may be due to the intrinsic differences between the SHSY5Y cell line and Ngn2 human neurons, or it may reflect the fact that editing in human neurons occurred during differentiation of the neurons. While Ngn2 neurons offered the ability to test cell type specific phenotypes in a pure population of excitatory glutamatergic neurons without having to account for the possibility of a diluted phenotype from mixed neuronal cultures^{42, 54}, the system has its limitations. At 14 days post differentiation, the iNgn2 neurons did express the 4R tau isoform and levels of total tau protein were not comparable to human brain levels. Previous studies have attempted to address this Ngn2 limitation by culturing neurons for longer. This increased the levels of 4R tau, but the 3R:4R Tau expression ratio was not comparable to expression ratios in human brains⁵⁵. Using an induced pluripotent stem cells (iPSC)-neuronal system from patients with tau mutations, differentiation for 150 to 365 days was required to see expression of multiple tau isoforms, yet these still did not compare to the levels seen in the human brains^{56, 57, 58}. While a more robust human neuronal system with expression of both 3R and 4R tau isoforms would benefit studies looking at tau aggregation, the scope of this study is only looking at regulators of tau protein levels. For these purposes the current neuronal system allowed for a rapid additional hit validation in a human post-mitotic neuronal system that expressed endogenous tau levels and triage of hits for validation in a more mature in-vivo neuronal system. Additional approaches such as CRISPR-i or

pharmacological inhibition in differentiated human neurons⁵⁵, or in vivo experiments may be required to validate additional genes and pathways that were identified in the SH-SY5Y screens.”

Reviewer Comments: Line 35 “OR” transition isn’t clear – Could be rephrased to OR [Line 38 - “secondary” where the tau lesion is downstream of a causative insult such as AB amyloid plaques in Alzheimer’s disease (AD)]. Another option is adding the word “when” following OR.

Response: Minor text edit has been addressed to make paragraph on Line 35 more clear.

Reviewer Comments: inaccurate wording using “only”- tau is not the ONLY aggregated protein found in brains of patients with PSP, CBD, PiD, and FTDP-17. Consider replacing with “predominant”.

Response: Minor text change replaced “only” with “predominant” to make statement more accurate.

Reviewer Comments: Lines 135, 159, 207, 208, and other instances: Font error (temperature unit).

Response: Minor text edit on all temperature units changes to proper readable symbol.

Reviewer Comments: Lines 573, 576, and other instances: brackets left open

Response: Reference page brackets left open have been all closed for reference page annotations.

Reviewer Comments: Line 594: If only the top 20 of 93 genes reducing and top 23 of 62 genes increasing tau levels were validated in human neurons, what was done with the remaining genes, and what did their bioinformatic evaluation reveal?

Response: Supplementary Table 6 has been included in the manuscript with a list of genes that were tested in human neurons and the remainder list of genes that have yet to be tested.

In addition 2 sentences have been added to the discussion to address the reviewers questions,

“Remaining genes to be tested in neurons that meet our stringent criteria include 36 genes downregulating, and 14 genes upregulating tau protein levels in SH-SY5Y cells. These remaining genes are represented in the different protein networks identified through STRING analysis.”

The analysis using STRING was done using the 93 and 62 genes that were validated in secondary mini pool screens, so the genes that remain to be tested in human neurons are represented in the network analysis in Figure 2 and Supplementary Figure 2.

Reviewer Comments: A brief paragraph reminding the reader of the link of TSC1 to the mTOR pathway would be helpful.

Response: A brief paragraph reminding the reader of TSC and the mTOR pathway has been added,

“TSC1 and TSC2 form the GTPase TSC complex that inhibits RAS homologue enriched in brain (RHEB). In normal cell conditions RHEB activates the mTOR pathway causing mRNA translation via phosphorylation of S6 kinase and the eukaryotic initiation factor 4E binding protein (4EBP1). Cellular homeostasis is maintained via Adenosine Monophosphate-activated Protein Kinase (AMPK) which inactivates the TSC complex leading to the activation of RHEB, therefore mutations inhibiting TSC1/2 lead to increases in mTOR activity^{43, 44.}”

Reviewers' Comments:

Reviewer #1:

Remarks to the Author:

The authors have adequately addressed my concerns. These screens are a valuable resource to the research community

Reviewer #2:

Remarks to the Author:

The authors extensively revised the manuscript according to the comments from reviewer, supplemented with new experiment data. The quality of the manuscript is much improved and can be accepted for publication.

Reviewer #3:

Remarks to the Author:

The responses and corrective action taken by the authors have addressed my concerns. I have no further requests.